# Structured In-context Environment Scaling for Large Language Model Reasoning

**Peng Yu, Zeyuan Zhao, Shao Zhang, Luoyi Fu, Xinbing Wang, Ying Wen**[*]
Shanghai Jiao Tong University
{pursuit_yp,zhaozeyuan1102,shaozhang,yiluofu,xwang8,ying.wen}@sjtu.edu.cn

## Abstract

Large language models (LLMs) have achieved significant advancements in reasoning capabilities through reinforcement learning (RL) via environmental exploration. As the intrinsic properties of the environment determine the abilities that LLMs can learn, the environment plays an important role in the RL finetuning process. An ideal LLM reasoning environment should possess three core characteristics: scalability, generalizable reasoning, and verifiability. However, existing mathematical and coding environments are difficult to scale due to heavy reliance on expert annotation, while the skills learned in game-based environments are too specialized to generalize. To bridge this gap, we introduce the **S**tructured **I**n-context **E**nvironment (SIE) framework. SIE achieves scalability by automatically constructing reasoning environments from large-scale structured data, where the rich compositional patterns naturally support generalizable reasoning. Moreover, the explicit schemas and reasoning chains in structured data provide a foundation for rule-based verifiability. Experimental results show that the SIE framework not only achieves substantial improvements in in-domain structured reasoning, but also enables the learned compositional reasoning skills to generalize effectively to out-of-domain mathematical and logical reasoning tasks. We further explored learning in information-limited partial SIEs and found that LLMs can infer the missing information through exploring the environment, leading to robust reasoning improvements and generalization performance. Our code can be available at https://github.com/PursuitYP/SIE_ICLR.

## 1 Introduction

Fine-tuning large language models (LLMs) with reinforcement learning (RL) has emerged as a dominant post-training paradigm for eliciting complex reasoning capabilities (Jaech et al., 2024; Guo et al., 2025; Team et al., 2025; Comanici et al., 2025). This mechanism of learning from environmental feedback enables LLMs to acquire crucial reasoning strategies such as self-reflection, backtracking, and chain-of-thought. RL fine-tuning has shown significant progress in math reasoning and code generation (Zeng et al., 2025; Hu et al., 2025b; Chen et al., 2025), and is gradually being extended to more challenging applications, such as interacting with search engines and building deep research agents (Jin et al., 2025; Zheng et al., 2025b; Li et al., 2025; Team, 2025).

Despite recent advancements in improving LLM reasoning via RL fine-tuning, existing research has focused primarily on algorithmic optimizations (Shao et al., 2024; Hu et al., 2025a; Zheng et al., 2025a), while the crucial role of the training environment has been comparatively overlooked. The intrinsic properties of the environment directly determine the capabilities that can be incentivized and shaped by the model. An ideal LLM reasoning environment should possess three key characteristics: (1) **Scalability**: The ability to construct large-scale, high-quality training environments from massive data sources in an automated and cost-effective manner. (2) **Generalizable Reasoning**: The reasoning strategies and cognitive patterns learned within the environment should be effectively transferred to other general-purpose reasoning domains. (3) **Verifiability**: The environment should possess clear, objective rules or mechanisms to verify the correctness of the answer.

---

[*]Ying Wen is the corresponding author.

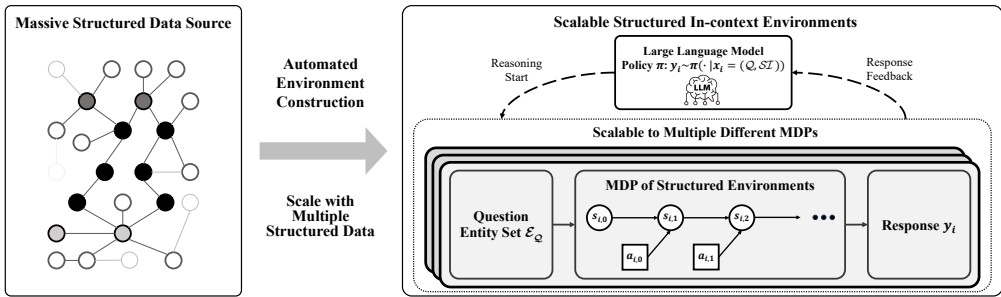

Figure 1: SIE constructs scalable, generalizable and verifiable in-context environments from structured data: an automated pipeline extracts local structured contexts from knowledge graphs, creates partial environments of varying difficulty, and uses rule-based reward to guide LLM learning.

A critical challenge in the current stage is how to automate the construction of scalable and high-quality LLM reasoning environments that meet the above requirements. However, existing LLM training environments generally fail to satisfy all these desiderata. One category is internalized-rule environments (e.g., mathematics), whose underlying structures are learned by LLMs during pre-training, but their construction relies on expensive expert annotations, limiting scalability (Cobbe et al., 2021; Lightman et al., 2023). Another category is externalized-rule environments (e.g., game engines), which have explicit rules, but the skills acquired from them are often highly specialized and do not generalize well to other reasoning domains (Wen et al., 2024; Zhang et al., 2025).

To address the challenges of high construction costs and limited generalization in existing RL environments, we explore the potential of automatically constructing such high-quality reasoning environments from massive structured data. Structured data refers to data organized according to a predefined schema, where fields, types, and constraints are explicitly defined, allowing for direct locating, retrieval, and querying of data items (Codd, 1970; Chang et al., 2019). Building training environments from structured data offers inherent advantages. First, the abundance of real-world structured resources (e.g., knowledge graphs and tabular data) enables automated and **scalable** environment construction through multi-hop retrieval and data composition. Second, since structured data represents a highly condensed form of human experience and domain knowledge, the reasoning patterns learned from it have strong potential to **generalize** to general reasoning tasks. Third, the explicit schemas and constraints inherent in structured data allow for rigorous rule-based **verification** of facts and outcomes. Therefore, building high-quality LLM training environments from structured data is not only feasible but also promising for balancing scalability and generalizability.

Motivated by these insights, we propose the **S**tructured **I**n-context **E**nvironment (SIE) framework. This framework is a flexible implementation of a structured environment, where its dynamics are encoded as a structured context and placed within the LLM's prompt as a soft constraint. The LLM's exploration within this context is modeled as implicit actions, and the resulting output can be directly used to derive reward signals for RL fine-tuning. This relaxed design simplifies implementation and scaling, while allowing seamless integration with mainstream RL fine-tuning algorithms. As shown in Figure 1, SIE comprises three core components: First, we design an automated pipeline to extract a local supportive structured environment from massive structured data to serve as the context for each task instance. Second, by dynamically controlling the effective information in this context, we construct a series of partial environments with varying difficulty to systematically study the learning efficiency and reasoning generalization of LLMs under information-constrained conditions. Finally, we devise a rule-based verifiable reward for RL fine-tuning to guide the LLM in learning the cognitive paradigms and compositional reasoning strategies embedded within the environment.

As a concrete implementation of the SIE framework, we choose knowledge graphs (KGs) as the structured data sources. KG triples provide a highly structured representation of human knowledge and contain domain-specific cognitive primitives; multi-hop paths formed by connecting multiple triples naturally correspond to complex reasoning processes and thus serve as excellent scaffolding for learning high-level compositional reasoning capability. We construct SIEs of varying scales and difficulties based on the Freebase KG (Bollacker et al., 2008) and fine-tune the Qwen and Llama series of models using the GRPO algorithm (Shao et al., 2024). Experimental results demonstrate that models fine-tuned with RL in the SIE not only achieve significant improvements on in-domain

structured reasoning tasks but also effectively transfer their learned reasoning strategies to out-of-domain mathematical and logical reasoning tasks, exhibiting superior generalization.

The main contributions of this paper are as follows:

- We propose and formalize the Structured In-context Environment (SIE) framework, using environmental complexity and context information as core experimental axes to systematically investigate the effectiveness and efficiency of fine-tuning LLMs with RL on SIEs.

- We automatically construct a series of partial SIEs of varying difficulty levels based on the Freebase KG. Experimental results not only validate the efficiency of RL fine-tuning in the constructed SIEs but also reveal that the learned cognitive pattern and compositional strategies can be generalized to boarder mathematical and logical reasoning domains.

- We provide a comprehensive analysis of how partial information affect LLM learning process, finding that information-constrained environments can effectively shift the model's reasoning paradigm from shallow memory retrieval to deeper compositional reasoning.

## 2 Structured In-context Environment for LLM Reasoning

This section presents the Structured In-context Environment (SIE) framework to improve the structured reasoning capabilities of LLMs and promote reasoning generalization. As shown in Figure 2, we first introduce how to automatically construct SIEs from large-scale KGs, and then explain how to treat SIEs as the in-context soft constraint to fine-tune LLMs with reinforcement learning (RL).

### 2.1 Construction Pipeline of SIEs

We instantiate the SIE framework using multi-hop knowledge graph question answering (KGQA) tasks and its underlying KGs. In KGQA tasks, the correct answer corresponds to a specific subgraph of KG $\mathcal{G}$ that contains the complete reasoning path from the question to the answer. Therefore, this subgraph serves as the ideal structured context for the task. As shown in Figure 1, the task is modeled as an implicit Markov Decision Process (MDP), where the LLM performs strategic exploration in the SIE based on the question. In the MDP, for the $i$-th sample at time step $t$, the state $s_{i,t}$ corresponds to the subgraph currently explored, the action $a_{i,t}$ corresponds to selecting the entity for further exploration, the state transition reflects the updated subgraph after executing the action, and final the reward $r_i$ is given by an external verifier based on the LLM response $y_i$. The automated SIE construction pipeline includes the following four steps: (1) seed subgraph retrieval, (2) supporting subgraph extraction, (3) distractor subgraph filtering, and (4) constructing partial SIEs.

**Step 1: Seed Subgraph Retrieval.** For each KGQA instance {question $\mathcal{Q}$, answer $\mathcal{A}$, question entity set $\mathcal{E}_{\mathcal{Q}}$, answer entity set $\mathcal{E}_{\mathcal{A}}$}, we treat the question entities in $\mathcal{E}_{\mathcal{Q}}$ as seed nodes and perform multi-hop retrieval on $\mathcal{G}$ to obtain an initial seed subgraph $\mathcal{G}_{seed}$ that contains potential reasoning paths. However, a naive breadth-first search would lead to exponential growth of the subgraph and severely impact processing efficiency. For example, a three-hop expansion from a single seed node in the Freebase KG, which contains 2.56 million entities and 8.3 million triples, can produce hundreds of thousands of triples. Thus, we adopt a more efficient bidirectional retrieval strategy: we perform multi-hop retrieval from both the question side and the answer side, while enforcing the sum of hops from the two directions equals the maximum hop $n_{hop}$ of the task. This approach significantly reduces the size of the seed subgraph and alleviates the computational burden for subsequent steps.

$$\mathcal{G}_{seed} = \text{MultiHopSearch}(\mathcal{G}, \mathcal{E}_{\mathcal{Q}}, q_{hop}) \cup \text{MultiHopSearch}(\mathcal{G}, \mathcal{E}_{\mathcal{A}}, a_{hop}), \quad (1)$$

where $\mathcal{G}$ is the original KG, $\mathcal{E}_{\mathcal{Q}}$ and $\mathcal{E}_{\mathcal{A}}$ are the sets of question and answer entities, respectively. The terms $q_{hop}$ and $a_{hop}$ represent the hop counts for the retrieval from the question and answer entities, where their sum must equal the maximum hop $n_{hop}$ of the task (i.e., $q_{hop} + a_{hop} = n_{hop}$).

**Step 2: Supporting Subgraph Extraction.** Given the seed subgraph $\mathcal{G}_{seed}$, our goal is to precisely extract all valid reasoning paths connecting the question entities $\mathcal{E}_{\mathcal{Q}}$ to the answer entities $\mathcal{E}_{\mathcal{A}}$, which together form the supporting subgraph $\mathcal{G}_{support}$. Considering that a question may involve multiple question entities and have multiple correct answers, we retain all question entities and the top ten correct answers. We then run the Dijkstra's algorithm to find all shortest paths between the source question entity set $\mathcal{E}_{\mathcal{Q}}$ and the target answer entity set $\mathcal{E}_{\mathcal{A}}$, within the maximum hop limit $n_{hop}$. The

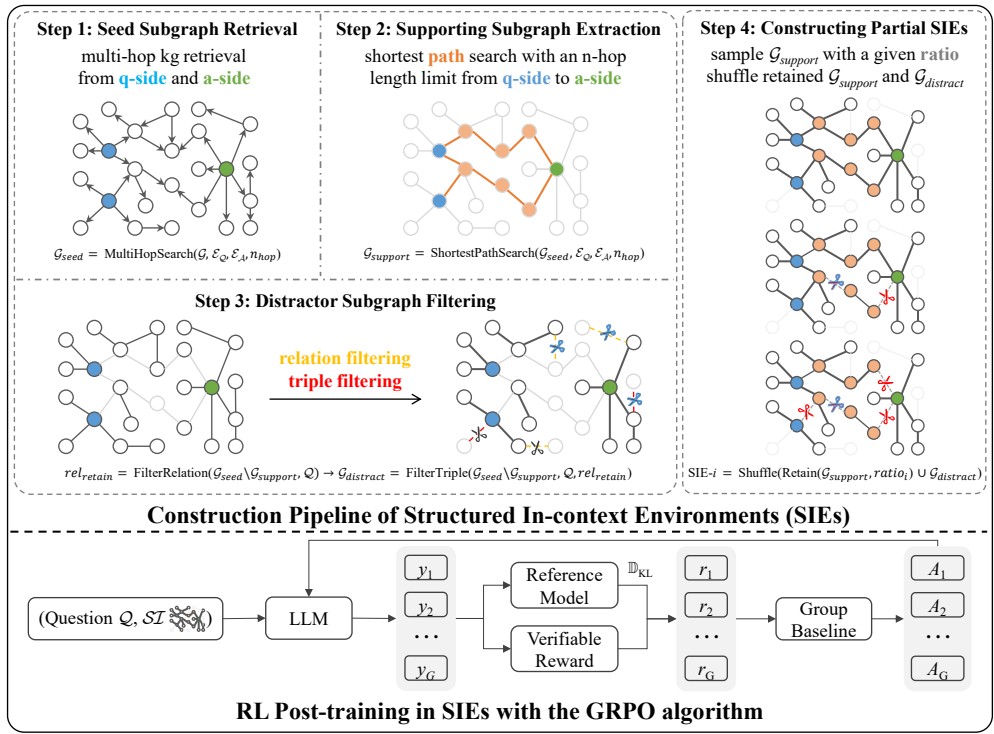

Figure 2: Overview of the **SIE** framework. **Up**: The automated construction pipeline for SIEs involves four key steps: (1) Seed Subgraph Retrieval; (2) Supporting Subgraph Extraction; (3) Distractor Subgraph Filtering; and (4) Constructing Partial SIEs. **Down**: We apply the GRPO algorithm to perform RL fine-tuning of LLMs within the SIEs to elicit structured reasoning capabilities.

resulting supporting subgraph $\mathcal{G}_{support}$ not only ensures the inclusion of the structured knowledge necessary to answer $\mathcal{Q}$ but also maintains a manageable size. Due to a semantic misalignment between $\mathcal{Q}$ and $\mathcal{G}$, the supporting subgraph for some questions may be empty; we retain these instances to study the impact of environmental incompleteness on the LLM reasoning and generalization.

$$\mathcal{G}_{support} = \text{ShortestPathSearch}(\mathcal{G}_{seed}, \mathcal{E}_Q, \mathcal{E}_A, n_{hop}), \tag{2}$$

where $\mathcal{G}_{seed}$ is the seed subgraph from the previous step and $n_{hop}$ is the maximum hop for the task.

**Step 3: Distractor Subgraph Filtering.** After removing the supporting subgraph $\mathcal{G}_{support}$ from the seed subgraph $\mathcal{G}_{seed}$, the remaining triples constitute the distractor subgraph $\mathcal{G}_{distract}$. However, the initial distractor subgraph is still too large (e.g., averaging nearly 10,000 triples), exceeding the context length limitations of LLMs. To resolve this, we designed a two-stage semantic filtering process to preserve the most relevant and challenging distractor information. Specifically, we use the pre-trained cross-encoder model ms-marco-MiniLM-L12-v2 for reranking. The first stage is relation filtering: we extract all relations from the initial distractor subgraph, calculate their semantic similarity to the original question $Q$, and retain the top-ranking relations $rel_{retain}$. The second stage is triple filtering: we keep only those triples with relation in $rel_{retain}$ from the previous step, and then calculate their semantic similarity to $Q$ and keep the top-ranking triples to form the final distractor subgraph $\mathcal{G}_{distract}$. This two-stage semantic ranking balances environment complexity design with context length constraints, producing $\mathcal{G}_{distract}$ that is meaningful and challenging.

$$rel_{retain} = \text{FilterRelation}(\mathcal{G}_{seed} \setminus \mathcal{G}_{support}, \mathcal{Q}), \tag{3}$$

$$\mathcal{G}_{distract} = \text{FilterTriple}(\mathcal{G}_{seed} \setminus \mathcal{G}_{support}, \mathcal{Q}, rel_{retain}), \tag{4}$$

where $\mathcal{G}_{seed}$ and $\mathcal{G}_{support}$ are the seed subgraph and supporting subgraph, respectively. The notation $\mathcal{G}_{seed} \setminus \mathcal{G}_{support}$ denotes the triples in $\mathcal{G}_{seed}$ that are not in $\mathcal{G}_{support}$, $\mathcal{Q}$ is the original question, and $rel_{retain}$ is the set of retained relations after the first-stage filtering.

**Step 4: Constructing Partial SIEs.** After completing the three subgraph extraction steps, we merge and randomly shuffle the triples from $\mathcal{G}_{support}$ and $\mathcal{G}_{distract}$ to form the final Structured In-context

Environment (SIE). Each sample in the SIE is represented as (question $\mathcal{Q}$, structured in-context $\mathcal{SI}$, answer $\mathcal{A}$), where the structured in-context $\mathcal{SI}$ is placed in the reasoning prompt to serve as a soft constraint. To systematically study the impact of varying difficulty and incomplete information on LLM reasoning, we constructed a series of partial SIEs by controlling the retention ratio of $\mathcal{G}_{support}$. Specifically, we set a series of retention ratios at $\{100\%, 75\%, 50\%, 25\%, 0\%\}$ and adjusted the size of $\mathcal{G}_{distract}$ accordingly to keep the total length of the context constant. This corresponds to five partial SIEs with increasing difficulty: SIE-100%, SIE-75%, SIE-50%, SIE-25%, and SIE-0%. This suite of SIEs simulates a progression from a complete to a progressively more incomplete environment, allowing us to systematically study how LLM reasoning evolve under information-constrained conditions.

$$\text{SIE-}ratio = \text{Shuffle}(\text{Retain}(\mathcal{G}_{support}, ratio) \cup \mathcal{G}_{distract}) \text{ for } ratio \in \{100\%, 75\%, 50\%, 25\%, 0\%\}, \quad (5)$$

where SIE-$ratio$ is the partial SIE for difficulty level $ratio$, $\mathcal{G}_{support}$ and $\mathcal{G}_{distract}$ are the supporting and distractor subgraphs, respectively. The function $\text{Retain}(\cdot, ratio)$ randomly samples a subset of the triples from $\mathcal{G}_{support}$ based on the corresponding retention $ratio$.

## 2.2 RL Post-training within SIEs

In the SIE framework, we treat the environment as a soft in-context constraint for LLM reasoning. The LLM is required to explore this provided in-context environment to perform multi-hop compositional reasoning. This setup makes it very convenient to fine-tune LLMs using various RL algorithms, which ensures training scalability. We leveraged the GRPO algorithm (Shao et al., 2024) to perform efficient RL fine-tuning on a range of open-source LLMs. This algorithm eliminates the need for a separate critic model and uses group relative scoring as a baseline to calculate the advantage, which significantly simplifies the training process. Given a question and its corresponding structured in-context as the reasoning input, denoted as $x = (\mathcal{Q}, \mathcal{SI})$, and a ground-truth answer $y^* = \mathcal{A}$ from the environment, GRPO samples a group of responses $\{y_1, y_2, \ldots, y_G\}$ from the old policy $\pi_{\theta_{old}}$ and optimizes the current policy model $\pi_\theta$ by maximizing the following objective:

$$\mathcal{J}_{\text{GRPO}}(\theta) = \mathbb{E}_{\substack{(x,y^*)\sim\mathcal{SIE} \\ \{y_i\}_{i=1}^G \sim \pi_{\theta_{old}}(\cdot|x)}} \Big[ \frac{1}{G} \sum_{i=1}^G \Big( \min\Big( \frac{\pi_\theta(y_i \mid x)}{\pi_{\theta_{old}}(y_i \mid x)} A_i, $$

$$\text{clip}\Big( \frac{\pi_\theta(y_i \mid x)}{\pi_{\theta_{old}}(y_i \mid x)}, 1-\epsilon, 1+\epsilon \Big) A_i \Big) - \beta\, \mathbb{D}_{\text{KL}}(\pi_\theta \,\|\, \pi_{\text{ref}}) \Big) \Big], \quad (6a)$$

$$\mathbb{D}_{\text{KL}}(\pi_\theta \,\|\, \pi_{\text{ref}}) = \frac{\pi_{\text{ref}}(y_i \mid x)}{\pi_\theta(y_i \mid x)} - \log \frac{\pi_{\text{ref}}(y_i \mid x)}{\pi_\theta(y_i \mid x)} - 1, \quad (6b)$$

$$A_i = \frac{r_i - \text{mean}(\{r_1, r_2, \ldots, r_G\})}{\text{std}(\{r_1, r_2, \ldots, r_G\})}, \quad (6c)$$

where $\epsilon$ and $\beta$ are hyper-parameters, and $A_i$ is the group-normalized advantage computed from the set of rewards $\{r_1, r_2, \ldots, r_G\}$ within each group.

For the structured reasoning template, we modified the DeepSeek-R1 (Guo et al., 2025) prompt to guide the model to perform step-by-step reasoning within `<think>` and `</think>` tags, placing the final answer in `<answer>` and `</answer>` tags. We used two types of rewards to perform RL fine-tuning on LLMs: an answer reward and a format reward. For the answer reward, we extract the final answer from the `<answer>` and `</answer>` tags and perform an exact match with the ground-truth answer, giving a reward of 1.0 for a successful match and 0.0 otherwise. For the format reward, we introduced an additional positive reward to encourage the model to follow the established thinking and answer paradigm. This rule-based reward mechanism effectively prevents reward hacking and ensures that the model optimizes toward the correct reasoning objective, guiding the LLM to learn the compositional reasoning paradigm inherent in the structured environment.

## 3 Experiments

To systematically evaluate the effectiveness of the SIE framework, we conducted comprehensive experiments to answer the following four research questions (RQs): (1) **RQ1**: Can using a structured environment as the context for LLM reasoning effectively elicit and improve structured reasoning capabilities? (2) **RQ2**: Compared to structured reasoning data (SRD), is the SIE more efficient in boosting the reasoning abilities of LLMs? (3) **RQ3**: Can the structured reasoning skills learned

within the SIE generalize to more general out-of-domain reasoning tasks? (4) **RQ4**: How does RL fine-tuning on partial SIEs affect the LLM's reasoning and generalization performance?

## 3.1 Experimental Setup

### 3.1.1 Datasets and Metrics

**Training Settings.** We constructed the SIE instances on the Freebase KG, leveraging the widely used KGQA datasets WebQSP (Yih et al., 2016) and CWQ (Talmor & Berant, 2018). Following the pipeline in Section 2.1, we constructed partial SIEs by adjusting the retention ratio of $\mathcal{G}_{support}$: SIE-100%, SIE-75%, SIE-50%, SIE-25%, and SIE-0%. This setup allows us to study how reasoning abilities evolve in information-constrained environments. In addition, we distill the structured contexts from SIE into the corresponding structured reasoning data (SRD) using the DeepSeek-R1 API (Guo et al., 2025), enabling a direct comparison of learning efficiency between SIE-based in-context RL fine-tuning and conventional supervised fine-tuning on structured data.

**Test Datasets.** For structured reasoning, we used the WebQSP, CWQ, and GrailQA (Gu et al., 2021) test sets to create similar SIEs for in-domain evaluation. Notably, GrailQA was held out from the training setting to serve as in-domain generalization. Following ToG (Sun et al., 2023), we randomly sample 1,000 samples from the original GrailQA test set for evaluation. For general reasoning evaluation, we conducted out-of-domain generalization tests in both the mathematical and logical reasoning domains. For mathematical reasoning, we used GSM8K (Cobbe et al., 2021) and MATH500 (Lightman et al., 2023), which stress arithmetic problem solving and higher-level symbolic/algebraic reasoning, respectively. For logical reasoning, we used two subsets of the Knights and Knaves puzzle dataset (Xie et al., 2024): KK-easy (simple scenarios with 2-3 characters) and KK-hard (complex scenarios with 4-5 characters). In the puzzle task, the model must deduce which characters are truth-telling knights and which are lying knaves based on a series of statements. For all datasets, we use strict zero-shot evaluation and report pass@1 performance as the metric.

### 3.1.2 Baselines

To comprehensively evaluate the effectiveness of the SIE framework, we used the following baseline setups: (1) **RL w/ SIE**: This is our proposed core framework, which involves using RL fine-tuning on LLMs within the series of constructed SIEs. (2) **CoT** (Chain-of-Thought Prompting): This is a training-free baseline that uses step-by-step prompting to guide the model to reason within the SIE environment and generate an answer. (3) **RL w/o Context**: This method removes the structured environment from the SIE, directly performing RL fine-tuning on the LLM using (question, answer) pairs. This baseline directly addresses RQ1 by verifying the effectiveness of SIEs for structured reasoning. (4) **SFT w/ SRD** (Supervised Fine-Tuning with Structured Reasoning Data): We used the DeepSeek-R1 API to convert samples from our constructed SIEs into corresponding Structured Reasoning Data (SRD) through chain-of-thought distillation and rejection sampling (Yuan et al., 2023). We then used supervised fine-tuning (SFT) to train the LLM on this SRD. For the SFT process, the LLM is prompted with (question, structured triples) and is required to generate the corresponding (reasoning chain, answer). This setup is designed to address RQ2 by investigating the training efficiency of RL fine-tuning in SIEs compared to conventional SFT training in SRD.

### 3.1.3 Implementation Details

We fine-tuned a variety of open-source LLMs, including Qwen2.5-7B-Instruct (Yang et al., 2025), Llama3.1-8B-Instruct (Grattafiori et al., 2024), Qwen2.5-7B, and Qwen3-8B, using the GRPO algorithm (Shao et al., 2024) within the constructed SIEs. Among these LLMs, Qwen2.5-7B-Instruct, Llama3.1-8B-Instruct, and Qwen3-8B are instruction-tuned models, while Qwen2.5-7B is a base model that has only undergone pre-training. The entire RL post-training pipeline within the SIE was implemented using the VeRL framework (Sheng et al., 2025). For all SIE instances (SIE-100%, SIE-75%, SIE-50%, SIE-25%, and SIE-0%), we used a maximum prompt length of 8,192 tokens and a maximum response length of 2,048 tokens. Unless specified otherwise, subsequent mention of SIE refers to the SIE-100% setting, which retains the complete supporting subgraph.

Table 1: Structured reasoning evaluation under different RL fine-tuning settings. The red number in parentheses indicates the performance gains of RL w/ SIE over RL w/o Context. RL within the SIE significantly surpasses RL without a structured context, demonstrating the **effectiveness** of SIE.

| Datasets | Qwen2.5-7B-Instruct | | Llama3.1-8B-Instruct | | Qwen2.5-7B | | Qwen3-8B | |
|---|---|---|---|---|---|---|---|---|
| | w/o Context | w/ SIE | w/o Context | w/ SIE | w/o Context | w/ SIE | w/o Context | w/ SIE |
| WebQSP | 59.7 | **93.4 (+33.7)** | 61.3 | 93.2 (+31.9) | 62.8 | 93.2 (+30.4) | 48.6 | 90.2 (+41.6) |
| CWQ | 36.7 | 87.7 (+51.0) | 39.7 | **89.7 (+50.0)** | 38.4 | 89.3 (+50.9) | 29.7 | 78.6 (+48.9) |
| GrailQA | 20.8 | **85.8 (+65.0)** | 24.9 | 85.0 (+60.1) | 19.5 | 81.5 (+62.0) | 21.8 | 85.1 (+63.3) |

Table 2: Structured reasoning evaluation results under different fine-tuning methods. The red numbers in parentheses indicate the performance gains of SFT w/ SRD and RL w/ SIE relative to CoT. RL fine-tuning in SIE significantly outperforms SFT on SRD, demonstrating the **efficiency** of SIE.

| Datasets | Qwen2.5-7B-Instruct | | | Llama3.1-8B-Instruct | | |
|---|---|---|---|---|---|---|
| | CoT | SFT w/ SRD | RL w/ SIE | CoT | SFT w/ SRD | RL w/ SIE |
| WebQSP | 26.3 | 40.5 (+14.2) | **93.4 (+67.1)** | 36.5 | 43.4 (+6.9) | 93.2 (+56.7) |
| CWQ | 34.4 | 43.3 (+8.9) | 87.7 (+53.3) | 37.2 | 49.5 (+12.3) | **89.7 (+52.5)** |
| GrailQA | 40.5 | 55.7 (+15.2) | **85.8 (+45.3)** | 43.6 | 60.0 (+16.4) | 85.0 (+41.4) |

## 3.2 MAIN RESULTS

**The SIE Framework Effectively Enhances LLM Structured Reasoning (RQ1).** To analyze the effectiveness of the structured environment, we compared two distinct RL fine-tuning baselines: RL w/o Context (no structured context provided) and RL w/ SIE (structured context provided). Table 1 summarizes the performance of various LLMs on three structured reasoning tasks: WebQSP, CWQ, and GrailQA. The results show a consistent and substantial performance improvement across all LLMs when RL fine-tuning is conducted within the SIE, compared to the setting without structured context. Specifically, after RL fine-tuning within the SIE, the LLMs achieved an average structured reasoning improvement of 34.4% on WebQSP, 50.2% on CWQ, and 62.6% on GrailQA. These results demonstrate the effectiveness of the SIE framework in promoting structured reasoning.

**RL Fine-tuning in SIE is More Efficient than SFT on SRD (RQ2).** Next, we analyzed the efficiency of SIE by comparing three reasoning baselines: CoT (Chain-of-Thought prompting), SFT w/ SRD (Supervised Fine-Tuning on Structured Reasoning Data), and RL w/ SIE (Reinforcement Learning fine-tuning in the Structured In-context Environment). Table 2 presents the results for Qwen2.5-7B-Instruct and Llama3.1-8B-Instruct across the three structured reasoning tasks. The results indicate that both SFT w/ SRD and RL w/ SIE yield consistent improvements over simple CoT prompting. Although LLMs fine-tuned by SFT w/ SRD achieved a modest average improvement of around 11.4% in structured reasoning across Qwen (11.3%) and Llama (11.5%) models, those fine-tuned by RL w/ SIE achieved a significantly greater average improvement of approximately 53.7% (55.6% for Qwen and 51.8% for Llama). Crucially, compared to the conventional SFT w/ SRD baseline, RL w/ SIE provided an additional performance gain exceeding 40% across all three structured reasoning tasks. These results demonstrate that RL fine-tuning within the SIE is more effective at encouraging environmental exploration and thus more efficiently improving the structured reasoning capabilities of LLMs than SFT imitation learning trained on the SRD.

**Structured Reasoning Learned in SIEs Generalizes to Out-of-Domain Reasoning Domains (RQ3).** We further analyzed the generalization of RL w/ SIE by evaluating performance on out-of-domain mathematical and logical reasoning tasks. Table 3 analyzes the performance of various LLMs on out-of-domain generalization datasets: GSM8K and MATH500 (representing simple and harder mathematical reasoning, respectively), and KK-easy (2-3 character logic puzzles) and KK-hard (4-5 character logic puzzles). Experimental results show that LLMs fine-tuned by RL w/ SIE achieve better generalization performance compared to CoT prmopting. Note that the lower initial accuracy of the Qwen3-8B model on the MATH500 task, compared to other LLMs, is attributed to the model frequently generating overly long responses or failing to adhere to the required reasoning format, resulting in a mismatch with the verifiable answer. This phenomenon is further analyzed in Appendix D. These LLMs achieved an average improvement of 20.4% on GSM8K, 18.1% on MATH500, 12.3% on KK-easy, and 11.1% on KK-hard after RL training. This indicates that the structured reasoning ability exhibits strong generalization to the math and logic reasoning domains.

**RL in Partial SIEs Achieves Robust Reasoning and Generalization Performance (RQ4).** Finally, we investigate the robustness of RL fine-tuning within the partial SIEs where environmental

Table 3: Out-of-domain reasoning generalization performance of different LLMs after RL fine-tuning in the in-domain SIEs. The red numbers in parentheses indicate the performance improvement of RL w/ SIE relative to CoT. These results demonstrate the strong generalizability of SIE.

| Datasets | Qwen2.5-7B-Instruct | | Llama3.1-8B-Instruct | | Qwen2.5-7B | | Qwen3-8B | |
|---|---|---|---|---|---|---|---|---|
| | CoT | RL w/ SIE | CoT | RL w/ SIE | CoT | RL w/ SIE | CoT | RL w/ SIE |
| GSM8K | 29.2 | 87.4 (+58.2) | 67.4 | 82.6 (+15.2) | 27.0 | 86.2 (+59.2) | 71.0 | **91.9 (+20.8)** |
| MATH500 | 43.0 | **61.6 (+18.6)** | 38.4 | 47.0 (+8.6) | 30.2 | 59.2 (+29.0) | 20.4 | 36.6 (+16.2) |
| KK-easy | 42.0 | 49.5 (+7.5) | 20.5 | 37.0 (+16.5) | 37.5 | 52.0 (+14.5) | 79.5 | **90.0 (+10.5)** |
| KK-hard | 19.5 | 29.0 (+9.5) | 6.0 | 15.5 (+9.5) | 15.5 | 27.5 (+12.0) | 59.5 | **73.5 (+14.0)** |

Table 4: Structured reasoning performance on WebQSP after RL fine-tuning in partial SIEs. The red numbers in parentheses indicate the performance improvement of RL w/ SIE relative to CoT.

| Setting | Qwen2.5-7B-Instruct | | Llama3.1-8B-Instruct | | Qwen2.5-7B | | Qwen3-8B | |
|---|---|---|---|---|---|---|---|---|
| | CoT | RL w/ SIE | CoT | RL w/ SIE | CoT | RL w/ SIE | CoT | RL w/ SIE |
| SIE-100% | 26.3 | **93.4 (+67.1)** | 36.5 | 93.2 (+56.7) | 2.6 | 93.2 (+90.6) | 47.8 | 90.2 (+42.4) |
| SIE-75% | 23.6 | 89.2 (+65.6) | 33.8 | **90.4 (+56.6)** | 2.0 | 90.2 (+88.2) | 47.3 | 88.0 (+40.7) |
| SIE-50% | 22.3 | 86.4 (+64.1) | 31.1 | **89.4 (+58.3)** | 2.5 | 87.2 (+84.7) | 44.9 | 84.0 (+39.1) |
| SIE-25% | 22.0 | 85.4 (+63.4) | 31.5 | **86.8 (+55.3)** | 1.6 | 85.8 (+84.2) | 44.8 | 82.6 (+37.8) |
| SIE-0% | 17.8 | 72.8 (+55.0) | 26.1 | **75.1 (+49.0)** | 1.7 | 73.4 (+71.7) | 36.7 | 70.8 (+34.1) |

information is incomplete. We compared five SIE settings, from SIE-100% to SIE-0%, which correspond to a gradually increasing difficulty in the structured environment. **Robustness of Structured Reasoning.** Table 4 compares the performance of various LLMs on the WebQSP structured reasoning task. All LLMs present a positive improvement in structured reasoning after RL fine-tuning across the five partial SIEs. In general, the performance of the LLMs gradually decreased as the environmental difficulty increased (from SIE-100% to SIE-0%), achieving average gains of 64.2%, 62.8%, 61.6%, 60.2%, and 52.5% respectively. Significantly, even in the most challenging scenarios, SIE-25% and SIE-0%, where only 25% of the supporting information was retained or all was removed, RL fine-tuning still brought about steady improvement in structured reasoning ability. This suggests that the reasoning pattern of LLMs evolved from shallow context retrieval to deeper compositional reasoning, learning to explore and compose information within the environment, leading to robust structured reasoning enhancement. **Robustness of Generalization.** Table 5 compares the generalization performance of Qwen2.5-7B-Instruct on the out-of-domain reasoning tasks. The LLM showed a positive improvement in general reasoning ability across all tasks after RL fine-tuning on partial SIEs. From SIE-100% to SIE-0%, the model achieved average performance gains in generalization of 40.3%, 40.1%, 38.6%, 38.6%, and 38.6% respectively. These results confirm that even when fine-tuned in partial SIEs, LLMs can robustly generalize the learned compositional structured reasoning abilities to general reasoning domains. As illustrated below, we selected a representative question from the SIE-0% scenario to compare the responses of LLMs before and after RL fine-tuning. Before RL fine-tuning, the model hallucinated and generated an incorrect answer. In contrast, after RL fine-tuning, the model recognized the insufficiency of the provided information and learned to combine its intrinsic parametric knowledge for complex multi-hop reasoning.

---

**Case study before and after RL fine-tuning in SIE**

**Question**: What is the Angelina Jolie directed movie that is 126 minutes long?

**Response before RL fine-tuning in SIE-0%**: From the knowledge graph, we know that Angelina Jolie directed the movie By the Sea and also directed A Place in Time...From the KG, it also states that the movie By the Sea is 126 minutes long...Thus, the movie directed by Angelina Jolie that is 126 minutes long is **By the Sea.** **[Wrong Answer!]**

**Response after RL fine-tuning in SIE-0%**: First, I need to find movies directed by Angelina Jolie...However, the knowledge triplets provided don't include the runtime information...I might have to rely on my own knowledge...Yes, In the Land of Blood and Honey **is a film directed by her and has a runtime of 126 minutes** (*learned to reasoning with its intrinsic knowledge*)...Therefore, the movie is **In the Land of Blood and Honey.** **[Correct Answer!]**

---

## 3.3 ANALYSIS

We conducted additional experiments to analyze the core characteristics of RL fine-tuning in SIEs. Specifically, we investigated: (1) the framework's applicability to mainstream RL algorithms; (2) its sensitivity to the RL starting checkpoint; (3) the impact of reward mechanisms to rule out format-driven gains; (4) the decomposition of performance sources across different environmental settings.

Table 5: Reasoning generalization performance of Qwen2.5-7B-Instruct after RL fine-tuning in partial SIEs. The red numbers in parentheses show the performance improvement relative to the initial CoT baseline after being trained with the corresponding RL w/ SIE.

| Setting | GSM8K | MATH500 | KK-easy | KK-hard |
|---------|-------|---------|---------|---------|
| CoT | 29.2 | 43.0 | 42.0 | 19.5 |
| SIE-100% | 87.4 (+58.2) | 61.6 (+18.6) | 49.5 (+7.5) | **29.0 (+9.5)** |
| SIE-75% | **87.7 (+58.5)** | 61.0 (+18.0) | **50.0 (+8.0)** | 26.0 (+6.5) |
| SIE-50% | 86.2 (+57.0) | 59.0 (+16.0) | 48.5 (+6.5) | 25.5 (+6.0) |
| SIE-25% | 86.0 (+56.8) | 60.2 (+17.2) | 48.0 (+6.0) | 24.5 (+5.0) |
| SIE-0% | 87.1 (+57.9) | 58.0 (+15.0) | 47.0 (+5.0) | 23.0 (+3.5) |

Table 6: Comparison of performance improvement in structured reasoning, mathematical reasoning, and logical reasoning tasks after fine-tuning Qwen2.5-7B-Instruct with different RL algorithms. The best results are highlighted in **bold**. REINFORCE++ and GRPO show comparable performance.

| Methods | WebQSP | CWQ | GrailQA | GSM8K | MATH500 | KK-easy | KK-hard |
|---------|--------|-----|---------|-------|---------|---------|---------|
| CoT | 26.3 | 34.4 | 40.5 | 29.2 | 43.0 | 42.0 | 19.5 |
| GRPO | **93.4** | 87.7 | **85.8** | **87.4** | 61.6 | **49.5** | **29.0** |
| REINFORCE++ | 93.1 | **88.4** | 83.2 | 86.7 | **62.2** | 49.0 | 24.5 |
| PPO | 85.4 | 73.4 | 81.4 | 78.4 | 59.6 | 49.0 | 25.0 |

Table 7: Comparison of performance improvement in structured, mathematical, and logical reasoning tasks after RL fine-tuning of Qwen2.5-7B-Instruct from different starting checkpoints. The best results are highlighted in **bold**. Compared to RL w/ SIE, RL w/ SIE f/ SFT achieves better generalization in math and logic reasoning, but its improvement in structured reasoning is limited.

| Methods | WebQSP | CWQ | GrailQA | GSM8K | MATH500 | KK-easy | KK-hard |
|---------|--------|-----|---------|-------|---------|---------|---------|
| SFT w/ SRD | 40.5 | 43.3 | 55.7 | 68.1 | 54.8 | 41.5 | 21.5 |
| RL w/ SIE | **93.4** | **87.7** | **85.8** | 87.4 | 61.6 | 49.5 | 29.0 |
| RL w/ SIE f/ SFT | 88.5 | 79.6 | 81.7 | **88.7** | **62.0** | **52.0** | **33.5** |

**The SIE Framework is Applicable to Mainstream RL Fine-tuning Algorithms.** We investigated the applicability of SIE by performing RL fine-tuning on Qwen2.5-7B-Instruct using REINFORCE++ (Hu et al., 2025a) and PPO (Schulman et al., 2017) algorithms in addition to GRPO. Table 6 summarizes the results in both the structured and general reasoning domains. The results indicate that the performance improvements and generalization achieved by REINFORCE++ are quite similar to the GRPO algorithm, while the gains from the PPO algorithm are comparatively weaker. All RL algorithms lead to improvements in structured reasoning capability and general reasoning ability. This demonstrates the universality of the SIE framework in RL fine-tuning algorithms.

**Starting RL from an SFT Checkpoint Enhances Generalization but Limits Structured Reasoning.** We investigated the effect of cold-starting RL training by using the model fine-tuned with SFT w/ SRD as a starting checkpoint for subsequent RL w/ SIE fine-tuning (labeled RL w/ SIE f/ SFT). Table 7 shows that RL w/ SIE f/ SFT leads to further gains in both structured and general reasoning compared to the SFT checkpoint itself. However, a comparison of RL w/ SIE f/ SFT and RL w/ SIE reveals a trade-off: the SFT-cold-started RL training performs worse on structured reasoning tasks (e.g., WebQSP: 88.5% vs. 93.4%), but achieves stronger generalization performance on out-of-domain tasks (e.g., KK-hard: 33.5% vs. 29.0%). These results suggest that the reasoning skills learned from the long-form responses in SRD can be more effectively generalized through RL refinement. However, the SFT cold-start constrains the LLM's ability to explore the environment, thereby limiting the maximum potential improvement in structured reasoning capability.

**Verifiable Environmental Feedback is Critical for Reasoning, Ruling Out Format Adherence and Reward Gaming.** To verify that the performance gains stem from learning correct reasoning logic rather than merely adhering to a specific output format or exploiting spurious signals, we introduced two ablation baselines under the RL w/ SIE-100% setting: *Format Only* (rewarding response structure without correctness) and *Random + Format* (replacing correctness reward with random 0-1 noise). Table 8 summarizes the results on Qwen2.5-7B-Instruct and Llama3.1-8B-Instruct. The results show that the *Format Only* baseline yields only marginal improvements over CoT (e.g., Qwen improves from 26.3% to 31.6% on WebQSP), primarily because standardized outputs facilitate answer extraction. However, this performance is significantly lower than the proposed *Answer*

Table 8: Ablation study on reward mechanisms. *Format Only* only rewards response structure without checking correctness, while *Random + Format* introduces random 0-1 noise. The significant gap between these baselines and the proposed *Answer + Format* reward function confirms that gains are driven by verifiable reasoning in SIEs, not format adherence or spurious correlations.

| Methods | WebQSP | CWQ | GrailQA | GSM8K | MATH500 | KK-easy | KK-hard |
|---|---|---|---|---|---|---|---|
| *Qwen2.5-7B-Instruct* | | | | | | | |
| + CoT | 26.3 | 34.4 | 40.5 | 29.2 | 43.0 | 42.0 | 19.5 |
| + Format Only | 31.6 | 37.7 | 48.1 | 37.0 | 46.6 | 44.0 | 20.0 |
| + Random + Format | 6.3 | 6.4 | 6.7 | 12.6 | 26.6 | 40.5 | 19.0 |
| + Answer + Format | **93.4** | **87.8** | **85.8** | **87.4** | **61.6** | **49.5** | **29.0** |
| *Llama3.1-8B-Instruct* | | | | | | | |
| + CoT | 36.5 | 37.2 | 43.6 | 67.4 | 38.4 | 20.5 | 6.0 |
| + Format Only | 45.1 | 44.1 | 56.1 | 68.8 | 43.4 | 30.0 | 11.0 |
| + Random + Format | 37.7 | 39.7 | 52.6 | 66.9 | 42.8 | 27.0 | 8.5 |
| + Answer + Format | **93.2** | **89.7** | **85.0** | **82.6** | **47.0** | **37.0** | **15.5** |

Table 9: Decomposition of performance gains across different environmental configurations. The step-wise improvements demonstrate how RL activates parametric knowledge, leverages negative constraints, and acieves compositional reasoning via internal and external knowledge synthesis.

| Datasets | CoT w/o Context | RL w/o Context | RL w/ SIE-0% | RL w/ SIE-100% |
|---|---|---|---|---|
| WebQSP | 2.0 | 59.7 | 72.8 | **93.4** |
| CWQ | 8.2 | 36.7 | 56.1 | **87.7** |

+ *Format* setting (93.4%), indicating that format adherence is not the primary driver of reasoning capability. Furthermore, in the *Random + Format* setting, the performance of the Qwen model collapses (dropping to ∼6%), while the Llama model also suffers significant degradation compared to the *Format Only* baseline. This demonstrates that the models are not gaming random signals; rather, the significant improvements in the SIE framework are driven by the model truly learning compositional reasoning patterns guided by verifiable structured environmental feedback.

**The SIE Framework Promotes Reasoning Evolution from Internal Knowledge Activation to Compositional Synthesis.** To deconstruct the sources of the structured reasoning improvements, we compared four progressive settings using Qwen2.5-7B-Instruct: CoT w/o Context, RL w/o Context, RL w/ SIE-0% (distractors only), and RL w/ SIE-100% (full context). In WebQSP and CWQ, approximately 65% and 40% of the questions are single-hop, respectively. Table 9 reveals a stepwise evolution in capability. First, the jump *from CoT w/o Context to RL w/o Context* (e.g., 2.0% → 59.7% on WebQSP) indicates that RL successfully activates the LLM's internal parametric knowledge, solving simpler, single-hop questions. Second, *comparing RL w/ SIE-0% to RL w/o Context* shows that even without supporting facts, the introduction of distractor subgraphs provides a negative constraint, boosting performance by an additional ∼13-20% by guiding the model to prune incorrect reasoning paths based on distractor subgraphs. Finally, the integration of supporting subgraphs in *RL w/ SIE-100% extends the knowledge boundary of LLMs*, yielding further ∼20-30% gain. This confirms that the complete SIE framework teaches the LLM to synthesize parametric knowledge with external structured evidence for complex, multi-hop compositional reasoning.

## 4 CONCLUSION

In this paper, we propose the SIE framework, which automatically constructs training environments for LLM reasoning from massive amounts of structured data. We further extended this by dynamically controlling the proportion of effective information in the structured in-context to build a series of partial SIEs for deeper analysis. We then performed RL fine-tuning on LLMs within these constructed SIEs to elicit their reasoning capabilities. Comprehensive experiments demonstrate that conducting RL fine-tuning within the SIE not only effectively boosts the structured reasoning abilities of LLMs but also generalizes significantly to more general out-of-domain reasoning tasks such as mathematics and logic. By analyzing the performance of LLMs trained in the partial SIEs, we found that RL fine-tuning efficiently encourages the model to explore the environment to infer missing information, leading to robust reasoning improvements and effective generalization.

**The Use of Large Language Models.** We used a large language model as a general-purpose assistant solely for text editing, including grammar correction, wording and tone adjustments, punctuation, and stylistic consistency. The model did not contribute to research ideation, methodology, experimental design, data analysis, interpretation of results, or the generation of substantive academic content or references. All suggestions were reviewed and approved by the authors, who take full responsibility for the final text.

**Ethics Statement.** Our method and algorithm do not involve any adversarial attack, and will not endanger human security. All our experiments are performed in the simulation environment, which does not involve ethical and fair issues.

ACKNOWLEDGMENTS

This work was supported by the National Key R&D Program of China (No. 2024YFC3505402), and the National Natural Science Foundation of China (No. U2244217 and No. 62525209).

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

# A    RELATED WORK

## A.1    IMPROVING LLM REASONING WITH RL

RL has significantly enhanced the reasoning capabilities of LLMs (Guo et al., 2025; Team et al., 2025; Xie et al., 2025). However, recent research on LLM reasoning has predominantly focused on the refinement and optimization of RL algorithms, with little attention paid to the importance of the RL environment itself (Shao et al., 2024; Hu et al., 2025a; Yu et al., 2025; Zheng et al., 2025a). Yet, the characteristics of the environment determine which specific capabilities of an LLM can be elicited. Specifically, environments based on mathematics and code focus on guiding general logical reasoning, but are difficult to scale due to their reliance on expensive expert annotations (Cobbe et al., 2021; Lightman et al., 2023). In contrast, game-based environments tend to cultivate task-oriented planning abilities, but the skills learned are often too specialized to generalize well (Carta et al., 2023; Tan et al., 2024; Wen et al., 2024). While concurrent work has begun to explore the construction of LLM reasoning environments from the perspectives of tool use, symbolic reasoning, and NP-hard graph problems (Fang et al., 2025; Lacombe et al., 2025; Wang et al., 2025), a formal definition of an ideal environment is lacking. An ideal LLM reasoning environment should possess three key attributes: scalability, generalizable reasoning, and verifiability. Therefore, we propose the automated construction of reasoning environments from structured data that satisfy these three attributes and the use of RL fine-tuning to efficiently elicit the reasoning capabilities of LLMs.

## A.2    PROMOTING LLM STRUCTURED REASONING

Despite notable advancements in mathematical and code reasoning (Zeng et al., 2025; Chen et al., 2025), LLMs still perform poorly on structured reasoning tasks that depend on external structured knowledge. Existing research to enhance the structured reasoning of LLMs falls mainly into two categories: task decomposition-based prompt engineering and supervised learning-based reasoning distillation. The former uses meticulously designed prompts to guide LLMs in exploring external knowledge bases with tools, gathering relevant structured knowledge to answer questions (Sun et al., 2023; Chen et al., 2024; Tan et al., 2025). The latter distills reasoning chains from supporting structured knowledge, using either rule-based methods or more powerful LLMs, and then enhances the structured reasoning abilities of LLMs through supervised fine-tuning (Luo et al., 2023; Wu et al., 2025; Dedhia et al., 2025). However, the reasoning skills learned through these methods are typically relatively specialized and rigid, struggling to generalize to dynamic structured reasoning domains. In light of this, we formulate structured reasoning tasks as a structured in-context environment and employ RL training to effectively elicit generalizable structured reasoning capabilities.

# B    MORE EXPERIMENTAL RESULTS

## B.1    FULL IN-DOMAIN AND OOD EVALUATIONS

We report the complete experimental results for four representative LLMs, Qwen2.5-7B-Instruct, Llama3.1-8B-Instruct, Qwen2.5-7B, and Qwen3-8B, across five partial SIE settings that retain 100%, 75%, 50%, 25%, and 0% of supporting triples (SIE-100% to SIE-0%). The fine-tuning approaches compared include a training-free Chain-of-Thought prompt (CoT), supervised fine-tuning on distilled structured reasoning data (SFT w/ SRD), and our environment-driven RL fine-tuning (RL w/ SIE); we also report the DeepSeek-R1 baseline. Table 10 summarizes performance and relative gains on structured reasoning (SIE-driven KGQA), while Table 11 shows out-of-domain (OOD) generalization gains on mathematical and logical reasoning benchmarks. The overall pattern is clear: *RL w/ SIE substantially outperforms both CoT and SFT w/ SRD across all SIE configurations, and even under the most information-scarce setting (SIE-0%) RL fine-tuning still yields meaningful improvements*. Although SFT w/ SRD can enhance long-form reasoning behaviors and sometimes aids cross-domain transfer, its aggregate gains are smaller than those achieved by in-context RL exploration. These results also illustrate the gradual degradation of performance as the structured in-context information is removed and highlight relative robustness differences among models, providing empirical support for the claim that SIE-driven RL encourages exploratory compositional reasoning under information constraints.

Table 10: Structured reasoning performance after RL fine-tuning in partial SIEs.

| Datasets | Settings | Qwen2.5-7B-Instruct | | | Llama3.1-8B-Instruct | | | LLM API |
| | | CoT | SFT w/ SRD | RL w/ SIE | CoT | SFT w/ SRD | RL w/ SIE | DeepSeek-R1 |
|---|---|---|---|---|---|---|---|---|
| **WebQSP** | SIE-100% | 26.3 | 40.5 (+14.2) | 93.4 (+67.1) | 36.5 | 43.4 (+6.9) | 93.2 (+56.7) | 86.3 |
| | SIE-75% | 23.6 | 38.9 (+15.3) | 89.2 (+65.6) | 33.8 | 43.1 (+9.3) | 90.4 (+56.6) | 85.6 |
| | SIE-50% | 22.3 | 36.7 (+14.4) | 86.4 (+64.1) | 31.1 | 40.8 (+9.7) | 89.4 (+58.3) | 83.3 |
| | SIE-25% | 22.0 | 36.9 (+14.9) | 85.4 (+63.4) | 31.5 | 40.0 (+8.5) | 86.8 (+55.3) | 83.6 |
| | SIE-0% | 17.8 | 28.2 (+10.4) | 72.8 (+55.0) | 26.1 | 34.6 (+8.5) | 75.1 (+49.0) | 78.1 |
| | w/o Context | 2.0 | 13.6 (+11.6) | 59.7 (+57.7) | 15.1 | 15.4 (+0.3) | 61.3 (+46.2) | 66.3 |
| **CWQ** | SIE-100% | 34.4 | 43.3 (+8.9) | 87.7 (+53.3) | 37.2 | 49.5 (+12.3) | 89.7 (+52.5) | 76.2 |
| | SIE-75% | 33.0 | 39.5 (+6.5) | 83.6 (+50.6) | 35.3 | 47.1 (+11.8) | 86.9 (+51.6) | 74.3 |
| | SIE-50% | 29.8 | 35.4 (+5.6) | 78.2 (+48.4) | 33.2 | 41.9 (+8.7) | 83.4 (+50.2) | 70.8 |
| | SIE-25% | 29.3 | 33.3 (+4.0) | 73.8 (+44.5) | 31.2 | 40.6 (+9.4) | 78.9 (+47.7) | 68.3 |
| | SIE-0% | 24.2 | 28.9 (+4.7) | 56.1 (+31.9) | 26.6 | 34.5 (+7.9) | 60.6 (+34.0) | 62.1 |
| | w/o Context | 8.2 | 15.5 (+7.3) | 36.7 (+28.5) | 14.8 | 18.0 (+3.2) | 39.7 (+24.9) | 46.7 |
| **GrailQA** | SIE-100% | 40.5 | 55.7 (+15.2) | 85.8 (+45.3) | 43.6 | 60.0 (+16.4) | 85.0 (+41.4) | 86.8 |
| | SIE-75% | 39.9 | 57.4 (+17.5) | 84.1 (+44.2) | 43.5 | 59.1 (+15.6) | 83.8 (+40.3) | 86.3 |
| | SIE-50% | 39.3 | 53.6 (+14.3) | 81.7 (+42.4) | 44.3 | 57.8 (+13.5) | 82.7 (+38.4) | 85.5 |
| | SIE-25% | 37.7 | 52.9 (+15.2) | 78.9 (+41.2) | 43.4 | 56.9 (+13.5) | 81.6 (+38.2) | 84.1 |
| | SIE-0% | 33.8 | 49.5 (+15.7) | 71.5 (+37.7) | 38.6 | 56.2 (+17.6) | 72.9 (+34.3) | 83.4 |
| | w/o Context | 1.9 | 6.9 (+5.0) | 20.8 (+18.9) | 5.9 | 9.2 (+3.3) | 24.9 (+19.0) | 37.8 |

| Datasets | Settings | Qwen2.5-7B | | | Qwen3-8B (Pretraining & Post-training) | | | LLM API |
| | | CoT | SFT w/ SRD | RL w/ SIE | CoT | SFT w/ SRD | RL w/ SIE | DeepSeek-R1 |
|---|---|---|---|---|---|---|---|---|
| **WebQSP** | SIE-100% | 2.6 | 39.8 (+37.2) | 93.2 (+90.6) | 47.8 | 43.6 (-4.2) | 90.2 (+42.4) | 86.3 |
| | SIE-75% | 2.0 | 38.3 (+36.3) | 90.2 (+88.2) | 47.3 | 42.0 (-5.3) | 88.0 (+40.7) | 85.6 |
| | SIE-50% | 2.5 | 36.8 (+34.3) | 87.2 (+84.7) | 44.9 | 42.3 (-2.6) | 84.0 (+39.1) | 83.3 |
| | SIE-25% | 1.6 | 36.9 (+35.3) | 85.8 (+84.2) | 44.8 | 41.9 (-2.9) | 82.6 (+37.8) | 83.6 |
| | SIE-0% | 1.7 | 29.2 (+27.5) | 73.4 (+71.7) | 36.7 | 35.3 (-1.4) | 70.8 (+34.1) | 78.1 |
| | w/o Context | 9.7 | 13.8 (+4.1) | 62.8 (+53.1) | 12.3 | 13.0 (+0.7) | 48.6 (+36.3) | 66.3 |
| **CWQ** | SIE-100% | 3.2 | 43.1 (+39.9) | 89.3 (+86.1) | 48.6 | 51.5 (+2.9) | 78.6 (+30.0) | 76.2 |
| | SIE-75% | 3.1 | 39.8 (+36.7) | 85.3 (+82.2) | 46.6 | 47.6 (+1.0) | 75.2 (+28.6) | 74.3 |
| | SIE-50% | 2.7 | 34.5 (+31.8) | 79.9 (+77.2) | 42.9 | 45.3 (+2.4) | 67.9 (+25.0) | 70.8 |
| | SIE-25% | 2.4 | 33.2 (+30.8) | 75.1 (+72.7) | 41.1 | 43.8 (+2.7) | 66.9 (+25.8) | 68.3 |
| | SIE-0% | 2.2 | 28.4 (+26.2) | 58.1 (+55.9) | 35.6 | 36.4 (+0.8) | 55.9 (+20.3) | 62.1 |
| | w/o Context | 11.4 | 15.6 (+4.2) | 38.4 (+27.0) | 16.8 | 16.3 (-0.5) | 29.7 (+12.9) | 46.7 |
| **GrailQA** | SIE-100% | 13.2 | 51.6 (+38.4) | 81.5 (+68.3) | 67.5 | 64.0 (-3.5) | 85.1 (+17.6) | 86.8 |
| | SIE-75% | 15.4 | 53.7 (+38.3) | 81.1 (+65.7) | 67.7 | 63.3 (-4.4) | 84.7 (+17.0) | 86.3 |
| | SIE-50% | 14.2 | 50.7 (+36.5) | 80.1 (+65.9) | 65.9 | 63.2 (-2.7) | 83.0 (+17.1) | 85.5 |
| | SIE-25% | 13.6 | 51.6 (+38.0) | 79.0 (+65.4) | 66.3 | 62.1 (-4.2) | 82.1 (+15.8) | 84.1 |
| | SIE-0% | 15.5 | 46.4 (+30.9) | 72.1 (+56.6) | 64.5 | 60.6 (-3.9) | 77.6 (+13.1) | 83.4 |
| | w/o Context | 3.4 | 6.3 (+2.9) | 19.5 (+16.1) | 10.6 | 8.7 (-1.9) | 21.8 (+11.2) | 37.8 |

Table 10 shows a consistent and striking pattern across models and KGQA benchmarks: RL fine-tuning within the SIE (RL w/ SIE) delivers far larger gains than either CoT prompting or supervised fine-tuning on distilled SRD. For Qwen2.5-7B-Instruct and Llama3.1-8B-Instruct, RL w/ SIE yields very high accuracy scores on WebQSP ($\sim$ 93.4 and 93.2 at SIE-100%), CWQ ($\sim$ 87.7 and 89.7), and GrailQA ($\sim$ 85.8 and 85.0), substantially outperforming SFT w/ SRD (which typically improves scores by $\sim 6 - 16$ points over CoT) and the CoT baseline itself. The gains produced by RL w/ SIE are also robust across the partial-SIE spectrum: although absolute accuracy declines as support triples are removed (SIE-100% $\rightarrow$ SIE-0%), RL w/ SIE maintains pronounced advantages even in the most information-scarce settings (e.g., WebQSP SIE-0%: RL still 72.8 for Qwen2.5-7B-Instruct vs. CoT 17.8). For Qwen2.5-7B and Qwen3-8B, a similar trend emerges: RL w/ SIE produces very large relative improvements (often raising weak CoT baselines into strong performance ranges), while SFT w/ SRD yields substantial but smaller improvements. The DeepSeek-R1 baseline generally sits between SFT and RL in absolute performance for many settings, illustrating that the SIE-driven KGQA task still poses a certain level of difficulty even for powerful LLMs. Overall, The results demonstrates that SIE-enabled RL exploration is a far more effective mechanism for eliciting high-quality structured reasoning than passive supervision or prompting alone.

Table 11 demonstrates that the compositional strategies learned via RL w/ SIE transfer strongly to out-of-domain math and logic tasks. For Qwen2.5-7B-Instruct, RL w/ SIE achieves $\sim$ 87.4 on GSM8K and $\sim$ 61.6 on MATH500 at SIE-100%, substantially exceeding SFT w/ SRD ($\sim$ 68.1 and 54.8) and CoT (29.2 and 43.0). This pattern holds across different partial SIE levels: RL w/ SIE maintains high GSM8K accuracy ($\sim 86 - 88$) and yields consistent improvements on MATH500 and the Knights & Knaves subsets (KK-easy, KK-hard). Llama3.1-8B-Instruct shows the same qualitative trend that RL w/ SIE improves GSM8K and logical-task performance over SFT, though absolute magnitudes vary by model and dataset. For Qwen2.5-7B and Qwen3-8B, RL w/ SIE similarly

Table 11: Out-of-domain generalization performance after RL fine-tuning in partial SIEs.

| | **Qwen2.5-7B-Instruct** | | | | | | | |
|---|---|---|---|---|---|---|---|---|
| | **GSM8K (29.2%)** | | **MATH500 (43.0%)** | | **KK-easy (42.0%)** | | **KK-hard (19.5%)** | |
| **Settings** | **SFT w/ SRD** | **RL w/ SIE** | **SFT w/ SRD** | **RL w/ SIE** | **SFT w/ SRD** | **RL w/ SIE** | **SFT w/ SRD** | **RL w/ SIE** |
| SIE-100% | 68.1 (+38.9) | 87.4 (+58.2) | 54.8 (+11.8) | 61.6 (+18.6) | 41.5 (-0.5) | 49.5 (+7.5) | 21.5 (+2.0) | 29.0 (+9.5) |
| SIE-75% | 63.3 (+34.1) | 87.7 (+58.5) | 54.0 (+11.0) | 61.0 (+18.0) | 39.5 (-2.5) | 50.0 (+8.0) | 24.5 (+5.0) | 26.0 (+6.5) |
| SIE-50% | 68.7 (+39.5) | 86.2 (+57.0) | 55.2 (+12.2) | 59.0 (+16.0) | 47.0 (+5.0) | 48.5 (+6.5) | 23.5 (+4.0) | 25.5 (+6.0) |
| SIE-25% | 63.9 (+34.7) | 86.0 (+56.8) | 52.0 (+9.0) | 60.2 (+17.2) | 46.0 (+4.0) | 48.0 (+6.0) | 28.5 (+9.0) | 24.5 (+5.0) |
| SIE-0% | 63.9 (+34.7) | 87.1 (+57.9) | 52.0 (+9.0) | 58.0 (+15.0) | 45.0 (+3.0) | 47.0 (+5.0) | 21.0 (+1.5) | 23.0 (+3.5) |
| w/o Context | 69.3 (+40.1) | 84.6 (+55.4) | 51.2 (+8.2) | 60.4 (+17.4) | 48.5 (+6.5) | 47.5 (+5.5) | 27.0 (+7.5) | 25.0 (+5.5) |

| | **Llama3.1-8B-Instruct** | | | | | | | |
|---|---|---|---|---|---|---|---|---|
| | **GSM8K (67.4%)** | | **MATH500 (38.4%)** | | **KK-easy (20.5%)** | | **KK-hard (6.0%)** | |
| **Settings** | **SFT w/ SRD** | **RL w/ SIE** | **SFT w/ SRD** | **RL w/ SIE** | **SFT w/ SRD** | **RL w/ SIE** | **SFT w/ SRD** | **RL w/ SIE** |
| SIE-100% | 73.6 (+6.2) | 82.6 (+15.2) | 42.0 (+3.6) | 47.0 (+8.6) | 8.5 (-12.0) | 37.0 (+16.5) | 1.5 (-4.5) | 15.5 (+9.5) |
| SIE-75% | 78.1 (+10.7) | 81.4 (+14.0) | 41.4 (+3.0) | 47.2 (+8.8) | 15.0 (-5.5) | 38.5 (+18.0) | 6.0 (+0.0) | 17.5 (+11.5) |
| SIE-50% | 75.2 (+7.8) | 81.7 (+14.3) | 40.4 (+2.0) | 46.4 (+8.0) | 13.0 (-7.5) | 35.0 (+14.5) | 1.0 (-5.0) | 14.0 (+8.0) |
| SIE-25% | 77.5 (+10.1) | 81.0 (+13.6) | 43.4 (+5.0) | 46.6 (+8.2) | 9.0 (-11.5) | 36.0 (+15.5) | 1.5 (-4.5) | 12.5 (+6.5) |
| SIE-0% | 77.1 (+9.7) | 81.2 (+13.8) | 41.8 (+3.4) | 45.8 (+7.4) | 10.5 (-10.0) | 38.5 (+18.0) | 2.0 (-4.0) | 14.5 (+8.5) |
| w/o Context | 75.1 (+7.7) | 77.2 (+9.8) | 44.8 (+6.4) | 43.4 (+5.0) | 25.0 (+4.5) | 35.5 (+15.0) | 5.0 (-1.0) | 12.5 (+6.5) |

| | **Qwen2.5-7B** | | | | | | | |
|---|---|---|---|---|---|---|---|---|
| | **GSM8K (27.0%)** | | **MATH500 (30.2%)** | | **KK-easy (37.5%)** | | **KK-hard (15.5%)** | |
| **Settings** | **SFT w/ SRD** | **RL w/ SIE** | **SFT w/ SRD** | **RL w/ SIE** | **SFT w/ SRD** | **RL w/ SIE** | **SFT w/ SRD** | **RL w/ SIE** |
| SIE-100% | 73.9 (+46.9) | 86.2 (+59.2) | 54.6 (+24.4) | 59.2 (+29.0) | 44.0 (+6.5) | 52.0 (+14.5) | 25.0 (+9.5) | 27.5 (+12.0) |
| SIE-75% | 71.8 (+44.8) | 86.6 (+59.6) | 52.6 (+22.4) | 57.4 (+27.2) | 39.5 (+2.0) | 51.5 (+14.0) | 25.0 (+9.5) | 26.0 (+10.5) |
| SIE-50% | 72.0 (+45.0) | 85.9 (+58.9) | 53.2 (+23.0) | 57.8 (+27.6) | 38.0 (+0.5) | 51.0 (+13.5) | 25.0 (+9.5) | 27.5 (+12.0) |
| SIE-25% | 68.8 (+41.8) | 87.7 (+60.7) | 51.2 (+21.0) | 58.8 (+28.6) | 37.0 (-0.5) | 51.5 (+14.0) | 24.5 (+9.0) | 29.5 (+14.0) |
| SIE-0% | 68.2 (+41.2) | 85.9 (+58.9) | 53.6 (+23.4) | 56.8 (+26.6) | 34.5 (-3.0) | 53.5 (+16.0) | 19.5 (+4.0) | 28.5 (+13.0) |
| w/o Context | 68.0 (+41.0) | 86.4 (+59.4) | 52.0 (+21.8) | 55.2 (+25.0) | 46.0 (+8.5) | 50.0 (+12.5) | 22.5 (+7.0) | 28.0 (+12.5) |

| | **Qwen3-8B (Pretraining & Post-training)** | | | | | | | |
|---|---|---|---|---|---|---|---|---|
| | **GSM8K (71.1%)** | | **MATH500 (20.4%)** | | **KK-easy (79.5%)** | | **KK-hard (59.5%)** | |
| **Settings** | **SFT w/ SRD** | **RL w/ SIE** | **SFT w/ SRD** | **RL w/ SIE** | **SFT w/ SRD** | **RL w/ SIE** | **SFT w/ SRD** | **RL w/ SIE** |
| SIE-100% | 78.4 (+7.3) | 91.9 (+20.8) | 40.8 (+20.4) | 36.6 (+16.2) | 83.0 (+3.5) | 90.0 (+10.5) | 66.0 (+6.5) | 73.5 (+14.0) |
| SIE-75% | 77.5 (+6.4) | 93.1 (+22.0) | 39.4 (+19.0) | 38.0 (+17.6) | 88.0 (+8.5) | 95.5 (+16.0) | 68.5 (+9.0) | 77.5 (+18.0) |
| SIE-50% | 78.6 (+7.5) | 89.4 (+18.3) | 40.4 (+20.0) | 36.8 (+16.4) | 84.5 (+5.0) | 89.0 (+9.5) | 67.0 (+7.5) | 73.0 (+13.5) |
| SIE-25% | 79.5 (+8.4) | 93.9 (+22.8) | 37.6 (+17.2) | 46.6 (+26.2) | 86.0 (+6.5) | 93.5 (+14.0) | 67.5 (+8.0) | 78.5 (+19.0) |
| SIE-0% | 79.1 (+8.0) | 93.4 (+22.3) | 40.6 (+20.2) | 44.6 (+24.2) | 85.5 (+6.0) | 94.5 (+15.0) | 64.5 (+5.0) | 80.0 (+20.5) |
| w/o Context | 83.5 (+12.4) | 90.2 (+19.1) | 40.6 (+20.2) | 35.7 (+15.3) | 94.0 (+14.5) | 89.5 (+10.0) | 72.5 (+13.0) | 67.5 (+8.0) |

produces strong OOD gains, often moving models from modest CoT baselines into substantially higher-performance regimes. Notably, SFT w/ SRD sometimes produces competitive or even superior results on certain math splits for particular models (reflecting that distilled long-form reasoning can benefit arithmetic tasks), but on average the RL w/ SIE condition yields larger and more consistent cross-domain gains. Together, the numbers indicate that SIE-driven RL induces compositional reasoning behaviors that generalize beyond the structured environment.

## B.2 GENERALIZATION ON HARD MATH AND TABULAR TASKS

**RL w/ SIE Demonstrates Strong Generalization on Olympiad-Level Math and Tabular Data.** To verify whether the learned strategies generalize to scarce-signal and highly challenging regimes, we evaluated our method on AIME 2024 (an Olympiad-level math benchmark) and TabMWP (Lu et al., 2022) (a table-based structured QA dataset). Table 12 reports the results for Qwen2.5-7B and Llama3.1-8B-Instruct. On AIME 2024, our method demonstrates stable performance advantages over the CoT baseline as $k$ increases (e.g., Qwen2.5-7B pass@8 improves from 9.22 to 19.41). This indicates that models trained within SIE possess stronger exploration capabilities and robustness when dealing with complex, multi-step reasoning tasks. Moreover, on TabMWP, our method achieves substantial performance improvements in the zero-shot setting (e.g., +36.3% for Qwen and +7.5% for Llama). This confirms that the reasoning capabilities cultivated by the SIE framework are not limited to KG structures but can effectively transfer to heterogeneous structured data like tables.

Table 12: Evaluation on AIME 2024 and TabMWP. RL w/ SIE significantly improves pass@k on the hard math benchmark and demonstrates strong zero-shot transfer capabilities to tabular data.

| Methods | AIME 24 pass@1 | AIME 24 pass@2 | AIME 24 pass@4 | AIME 24 pass@8 | TabMWP accuracy |
|---|---|---|---|---|---|
| *Qwen2.5-7B (Base)* | | | | | |
| + CoT | 2.29 | 3.97 | 6.27 | 9.22 | 45.5 |
| + RL w/ SIE-100% | **6.25** | **10.31** | **15.02** | **19.41** | **81.8** |
| *Llama3.1-8B-Instruct* | | | | | |
| + CoT | 3.12 | 4.97 | 7.31 | 10.89 | 69.5 |
| + RL w/ SIE-100% | **4.58** | **7.81** | **12.12** | **17.30** | **77.0** |

Table 13: Ablation study on different reranking strategies for distractor subgraphs. The semantic reranker provides the optimal trade-off between in-domain performance and OOD generalization.

| Methods | WebQSP | CWQ | GrailQA | GSM8K | MATH500 | KK-easy | KK-hard |
|---|---|---|---|---|---|---|---|
| Semantic Reranker | 93.4 | 87.8 | **85.8** | **87.4** | **61.6** | **49.5** | **29.0** |
| Random Reranker | 93.2 | 87.6 | 84.6 | 87.0 | 61.4 | 49.5 | 26.5 |
| Structure Reranker | **94.9** | **91.2** | 83.8 | 87.1 | 60.2 | 47.0 | 24.5 |

## B.3 Ablation Study on Distractor Rerankers

**The Semantic Reranker Balances Difficulty and Generalization, while Structural Similarity Leads to Shortcut Learning.** To verify the necessity and safety of our semantic reranking strategy, we compared it with two baselines: *Random Reranker* (randomly retaining distractor triples) and *Structure Reranker* (retaining triples selected through rule-based heuristics that prioritize structural similarity to the supporting subgraph or the presence of entity or relation mentions.). All experiments were conducted with the Qwen2.5-7B-Instruct + RL w/ SIE-100% setting. As shown in Table 13, the *Semantic Reranker* achieves the best overall performance, particularly in terms of generalization. While the *Random Reranker* yields comparable results on most tasks, it exhibits a notable decline on the challenging KK-hard logic benchmark (26.5% vs. 29.0%), suggesting that random distractors may lack sufficient relevance to establish a challenging reasoning boundary. Conversely, the *Structure Reranker* achieves the highest in-domain scores (e.g., 94.9% on WebQSP) but suffers from the poorest generalization (e.g., dropping to 24.5% on KK-hard). This suggests that overly structure-similar distractors can push the model to rely on superficial structural shortcuts instead of cultivating true exploration abilities, ultimately impairing its generalization.

## B.4 Scalability to Larger Models

**The SIE Framework Scales Effectively to Larger Model Sizes.** To investigate the scalability of our approach, we applied the SIE framework to the larger *Qwen2.5-14B-Instruct* model and compared it with the 7B version. Table 14 demonstrates that the 14B model achieves superior results under the RL w/ SIE-100% setting across all in-domain and out-of-domain tasks compared to the 7B model (e.g., MATH500 improves from 61.6% to 75.0%, and KK-hard improves from 29.0% to 45.5%). These consistent improvements confirm that the SIE framework is not limited to smaller models but can effectively scale to enhance the reasoning capabilities of larger foundational models.

## B.5 Comparison with Tool-using Agents

**RL w/ SIE Internalizes Reasoning Capabilities, Outperforming Tool-using Agents on Small Models.** We compared our method with *Think-on-Graph (ToG)* (Sun et al., 2023), a representative tool-using agent approach that utilizes structured data as external tools and context. As shown in Table 15, the ToG method relies heavily on the model's intrinsic instruction-following and planning capabilities. While it performs well with GPT-3.5 and GPT-4, it fails significantly with 7B-scale models (e.g., Qwen2.5-7B-Instruct + ToG achieves only 32.1% on WebQSP). In contrast, our RL w/ SIE method enables the 7B model to achieve a qualitative leap in structured reasoning. Remarkably, even in partial environments like *SIE-50%* (retaining only 50% supporting facts) or *SIE-0%* (no

Table 14: Comparison of performance between Qwen2.5-7B-Instruct and Qwen2.5-14B-Instruct. The 14B model achieves consistent gains, demonstrating the scalability of the SIE framework.

| Methods | WebQSP | CWQ | GrailQA | GSM8K | MATH500 | KK-easy | KK-hard |
|---|---|---|---|---|---|---|---|
| *Qwen2.5-7B-Instruct* | | | | | | | |
| + CoT | 26.3 | 34.4 | 40.5 | 29.2 | 43.0 | 42.0 | 19.5 |
| + RL w/ SIE-100% | 93.4 | 87.8 | 85.8 | 87.4 | 61.6 | 49.5 | 29.0 |
| *Qwen2.5-14B-Instruct* | | | | | | | |
| + CoT | 40.9 | 48.0 | 65.6 | 72.1 | 62.4 | 60.5 | 35.0 |
| + RL w/ SIE-100% | **94.0** | **89.9** | **87.4** | **91.1** | **75.0** | **66.0** | **45.5** |

Table 15: Comparison with Tool-using Agents. The results for GPT-3.5+ToG and GPT-4+ToG are taken from the original ToG paper. RL w/ SIE significantly outperforms the ToG agent on 7B models and matches GPT-3.5+ToG performance even under information-limited partial SIEs.

| Methods | WebQSP | CWQ | GrailQA |
|---|---|---|---|
| *Qwen2.5-7B-Instruct* | | | |
| + SIE-100% + CoT | 26.3 | 34.4 | 40.5 |
| + RL w/ SIE-100% | **93.4** | **87.8** | **85.8** |
| + RL w/ SIE-50% | 86.4 | 78.2 | 81.7 |
| + RL w/ SIE-0% | 72.8 | 56.1 | 71.5 |
| + ToG | 32.1 | 26.0 | 15.3 |
| *Closed-source Models* | | | |
| GPT-3.5 + ToG | 76.2 | 57.1 | 68.7 |
| GPT-4 + ToG | 82.6 | 67.6 | 81.4 |

supporting facts), the RL-trained 7B model outperforms or matches the much larger GPT-3.5 + ToG baseline. This compelling evidence demonstrates that the SIE framework transcends conventional context augmentation strategies. Rather than passively relying on retrieved knowledge, it utilizes RL to deeply internalize the complex heuristics of structured exploration and multi-hop logical deduction directly into the model's parameters. By transforming these external reasoning trajectories into intrinsic cognitive capabilities, SIE effectively overcomes the inherent planning bottlenecks of smaller models, empowering them to autonomously perform complex structured reasoning.

## C PROMPTS

The reasoning prompt for the SIE-based KGQA task is shown below:

---

**Structured Reasoning Prompt for the SIE Framework**

You are a structured reasoning expert tasked with solving problems step by step.
You first think about the reasoning process in the mind and then provide the answer.
The reasoning process and answer should be enclosed within `<think> </think>` and `<answer> </answer>` tags, respectively, i.e., `<think>` reasoning process here `</think>` `<answer>` answer here `</answer>`.
Given a question and the associated knowledge graph triplets (entity, relation, entity), you are asked to answer the question with these triplets and your own knowledge.
You can break down the original question into several sub-questions and explore the given knowledge graph to solve them step-by-step.
If there are multiple possible answers, choose one as the final answer.
After thinking, when you finally reach a conclusion, clearly state the answer within `<answer> </answer>` tags, i.e., `<answer>` answer here `</answer>`.

**Question**: {*question*}
**Knowledge Triplets**: {*knowledge triples*}

---

# D CASE STUDIES

We present case studies that compare the behavior of LLMs before and after RL fine-tuning in the SIE, and we observe that Qwen3-8B has a tendency to produce excessively lengthy responses.

---

**Case study before and after RL fine-tuning in SIE**

**Question**: What is the Angelina Jolie directed movie that is 126 minutes long?

**Response before RL fine-tuning in SIE-0%**:

```
<think>
```
The question asks for a movie directed by Angelina Jolie that is 126 minutes long. From the knowledge graph, we know that Angelina Jolie directed the movie By the Sea and also directed A Place in Time. Another piece of information is that Angelina Jolie produced the movie Unbroken. Specifically, we are looking for a movie that is 126 minutes long. From the knowledge graph, it also states that the movie By the Sea is 126 minutes long. Therefore, we need to confirm if there are any other movies directed by Angelina Jolie that are also 126 minutes long. From the knowledge graph, there are no other movies directed by Angelina Jolie that are 126 minutes long. Thus, the movie directed by Angelina Jolie that is 126 minutes long is By the Sea.
```
</think>
<answer>
```
By the Sea    **[Wrong Answer!]**
```
</answer>
```

**Response after RL fine-tuning in SIE-0%**:

```
<think>
```
Okay, let's tackle this question step by step. The question is asking for a movie directed by Angelina Jolie that is 126 minutes long. First, I need to parse the question. The key parts here are: the movie must be directed by Angelina Jolie, and its runtime is 126 minutes. Looking at the knowledge triplets provided, I need to find movies directed by Angelina Jolie. Let's scan through the triplets for any that mention Angelina Jolie as a director. I see a few entries like Angelina Jolie, film.director.film, Unbroken and Angelina Jolie, film.director.film, A Place in Time and By the Sea, film.film.directed_by, Angelina Jolie. So, the movies directed by her are Unbroken, A Place in Time, By the Sea, and possibly others. Now, I need to find the runtime of these movies. However, the knowledge triplets provided don't include the runtime information. Wait, but maybe there's another way. The question is asking for a movie that is 126 minutes long. I might need to recall some film runtimes. Unbroken is a movie directed by Angelina Jolie. I remember that Unbroken has a runtime of approximately 127 minutes. That's close to 126, but not exact. Then there's By the Sea, which I think is around 145 minutes. A Place in Time might be shorter, maybe around 100 minutes. Wait, but the question specifies 126 minutes. Since the knowledge triplets don't provide the runtime, I might have to rely on my own knowledge here. Alternatively, maybe there's another movie directed by her that I'm forgetting. Another movie directed by Angelina Jolie is In the Land of Blood and Honey, which I believe has a runtime of 126 minutes. Let me confirm that. Yes, In the Land of Blood and Honey is a film directed by her and has a runtime of 126 minutes. So, the answer would be In the Land of Blood and Honey. Let me check if that movie is listed in the knowledge triplets. Looking through the triplets, I see In the Land of Blood and Honey, film.film.directed_by, Angelina Jolie and In the Land of Blood and Honey, film.film.produced_by, Angelina Jolie. So, that's a valid entry. Therefore, the movie is In the Land of Blood and Honey.
```
</think>
<answer>
```
In the Land of Blood and Honey    **[Correct Answer!]**
```
</answer>
```

---

**Qwen3-8B often generates overly long responses**

**Question**: Find the product of the $y$-coordinates of all the distinct solutions $(x, y)$ for the two equations $y = x^2 - 8$ and $y^2 = -5x + 44$.

**Response of Qwen3-8B**:

```
<think>
```
Okay, so I need to find the product of the $y$-coordinates of all the distinct solutions $(x, y)$ for the two equations $y = x^2 - 8$ and $y^2 = -5x + 44$. Hmm, let me think about how to approach this.

First, since both equations involve $y$ and $x$, maybe I can substitute one into the other. The first equation gives $y$ in terms of $x$, so maybe I can plug that expression for $y$ into the second equation. Let me try that.

So, substituting $y = x^2 - 8$ into the second equation $y^2 = -5x + 44$, I get:

$$(x^2 - 8)^2 = -5x + 44.$$

Now, let me expand the left side. $(x^2 - 8)^2$ is $x^4 - 16x^2 + 64$. Wait, let me check that again. $(a - b)^2$ is $a^2 - 2ab + b^2$, so here $a$ is $x^2$ and $b$ is 8. So, $(x^2)^2$ is $x^4$, then $-2(x^2)(8)$ is $-16x^2$, and then $+8^2$ which is 64. Yes, so the left side is $x^4 - 16x^2 + 64$. So the equation becomes:

$$x^4 - 16x^2 + 64 = -5x + 44.$$

Let me bring all terms to one side to set the equation to zero:

$$x^4 - 16x^2 + 64 + 5x - 44 = 0.$$

Simplify the constants: $64 - 44$ is 20. So:

$$x^4 - 16x^2 + 5x + 20 = 0.$$

Hmm, so now I have a quartic equation: $x^4 - 16x^2 + 5x + 20 = 0$. Solving quartic equations can be complicated, but maybe this factors? Let me try to factor it.

First, let me check for rational roots using the Rational Root Theorem. The possible rational roots are factors of the constant term (20) divided by factors of the leading coefficient (1), so possible roots are $\pm 1, \pm 2, \pm 4, \pm 5, \pm 10, \pm 20$.

Let me test $x = 1$: $1 - 16 + 5 + 20 = 1 - 16$ is $-15$, $+5$ is $-10$, $+20$ is 10. Not zero.

$x = -1$: $1 - 16(-1)^2 + 5(-1) + 20$. Wait, $(-1)^2$ is 1, so $1 - 16 + (-5) + 20 = 1 - 16$ is $-15$, $-5$ is $-20$, $+20$ is 0. Oh! $x = -1$ is a root. Great!

So, $(x + 1)$ is a factor. Let me perform polynomial division or use synthetic division to factor it out.

Using synthetic division with root $-1$:

Coefficients of the polynomial $x^4 + 0x^3 - 16x^2 + 5x + 20$. Wait, the original polynomial is $x^4 - 16x^2 + 5x + 20$. So, the coefficients are $1$ ($x^4$), $0$ ($x^3$), $-16$ ($x^2$), $5$ ($x$), $20$ (constant).

Set up synthetic division:

$$
\begin{array}{r|rrrrr}
-1 & 1 & 0 & -16 & 5 & 20 \\
& & -1 & 1 & 15 & -20 \\
\hline
& 1 & -1 & -15 & 20 & 0
\end{array}
$$

So, the quotient polynomial is $x^3 - x^2 - 15x + 20$. Therefore, the original polynomial factors as $(x+1)(x^3 - x^2 - 15x + 20)$.

Now, let's try to factor the cubic polynomial $x^3 - x^2 - 15x + 20$. Again, use the Rational Root Theorem. Possible roots are $\pm 1, \pm 2, \pm 4, \pm 5, \pm 10, \pm 20$.

Test $x = 1$: $1 - 1 - 15 + 20 = 5 \neq 0$.

$x = 2$: $8 - 4 - 30 + 20 = (8 - 4) = 4, (4 - 30) = -26, (-26 + 20) = -6 \neq 0$.

$x = 5$: $125 - 25 - 75 + 20 = 125 - 25$ is $100$, $100 - 75$ is $25$, $25 + 20$ is $45 \neq 0$.

$x = 4$: $64 - 16 - 60 + 20 = 64 - 16$ is $48$, $48 - 60$ is $-12$, $-12 + 20$ is $8 \neq 0$.

$x = -1$: $-1 - 1 + 15 + 20 = 33 \neq 0$.

$x = 2$ again? Wait, already tried. How about $x = 5$? Wait, tried that. Maybe $x = 10$? Probably too big. Let me check $x = 2$ again. Wait, maybe I made a mistake.

Wait, $x = 2$: $8 - 4 - 30 + 20 = (8 - 4) = 4$; $(4 - 30) = -26$; $(-26 + 20) = -6$. Correct. Not zero.

$x = -2$: $-8 - 4 + 30 + 20 = (-12) + 50 = 38 \neq 0$.

$x = 5$? Wait, tried that. How about $x = 4$? $64 - 16 - 60 + 20 = 8$. Not zero. Maybe $x = 5/$ something? Wait, maybe I need to try another approach. Maybe factor by grouping?

Looking at $x^3 - x^2 - 15x + 20$. Let me group terms:

$$(x^3 - x^2) + (-15x + 20) = x^2(x - 1) - 5(3x - 4).$$

Hmm, not helpful. Alternatively, maybe different grouping?

Alternatively, maybe there's a typo? Wait, let me check my synthetic division again. Original polynomial was $x^4 - 16x^2 + 5x + 20$. When I divided by $(x + 1)$, the coefficients were $1, 0, -16, 5, 20$. Synthetic division steps:

Bring down $1$. Multiply by $-1$: $-1$. Add to next coefficient: $0 + (-1) = -1$. Multiply by $-1$: $1$. Add to next coefficient: $-16 + 1 = -15$. Multiply by $-1$: $15$. Add to next coefficient: $5 + 15 = 20$. Multiply by $-1$: $-20$. Add to last coefficient: $20 + (-20) = 0$. Correct. So the quotient is $x^3 - x^2 - 15x + 20$. So that's correct.

Alternatively, maybe there's another rational root. Let me check $x = 5$ again. Wait, $x = 5$: $125 - 25 - 75 + 20 = 125 - 25$ is $100$, $100 - 75$ is $25$, $25 + 20$ is $45$. Not zero. How about $x = 4$? $64 - 16 - 60 + 20 = 8$. Not zero. $x = 3$? $27 - 9 - 45 + 20 = (27 - 9) = 18$, $(18 - 45) = -27$, $(-27 + 20) = -7$. Not zero. $x = 2.5$? Maybe not rational.

Alternatively, maybe I made a mistake in the earlier steps?

Wait, let me check if there's another root. Maybe $x = -2$? Let me check $x = -2$ in the cubic: $(-8) - 4 - (-30) + 20 = -8 - 4 + 30 + 20 = 38$. Not zero. Hmm. Maybe I need to use the rational root theorem again. Wait, maybe I missed a possible root? Let me check $x = 5$ again. No. How about $x = 10$? $1000 - 100 - 150 + 20 = 1000 - 100$ is $900$, $900 - 150$ is $750$, $750 + 20$ is $770$. Not zero. Maybe $x = -4$? $(-64) - 16 - (-60) + 20 = -64 - 16 + 60 + 20 = (-80) + 80 = 0$! Oh! $x = -4$ is a root!

Wait, let me check that again. If $x = -4$, then:

$$x^3 - x^2 - 15x + 20 = (-4)^3 - (-4)^2 - 15(-4) + 20 = -64 - 1$$

[The output was truncated because it exceeded the context length!]

