# OpenReview forum: "Structured In-context Environment Scaling for Large Language Model Reasoning"
_ICLR.cc/2026/Conference — ICLR 2026 Poster_

### Official Review · Reviewer_Y1pk · 2025-10-24

**Soundness:** 3
**Presentation:** 3
**Contribution:** 2
**Rating:** 4
**Confidence:** 4

**Summary:**

This paper proposes a reinforcement learning framework that trains large language models to reason within automatically constructed Structured In-context Environments (SIEs). Each SIE is a mini structured world built from knowledge graphs, containing supporting and distractor facts that the model must navigate to answer questions. The authors also introduce a Structured Reasoning Dataset (SRD)—a supervised corpus generated from the same environments—to warm up the model before RL. Experiments on KGQA, math, and logic reasoning show that RL in SIEs substantially improves compositional reasoning and robustness to incomplete information.

**Strengths:**

The paper presents a clear pipeline for automatically generating verifiable reasoning environments via embedding the structural graphs into the context of LLMs. The idea is intuitive and the results demonstrate strong gains on structured reasoning benchmarks and generalization to math and logic tasks.

**Weaknesses:**

The idea of embedding structured as reasoning context is interesting but not conceptually new. I believe there are lots of related works on representing the structural data in the context of LLMs. It would be better for the authors to discuss them.

**Questions:**

See Weaknesses

---

> ### Author Response · Authors · 2025-11-22
> **Response to Reviewer Y1pk**
>
> We thank the reviewer for their valuable comments. We fully agree with your observation that representing structured data in the context of LLMs is a well-explored area. However, we would like to clarify that the core contribution of this paper is not merely using structured data as static context input, but rather **proposing a framework to automatically construct scalable, verifiable, and generalizable Reinforcement Learning environments (SIEs) from massive structured data**.
>
> Existing works on structured reasoning (as **discussed in Appendix A Related Work**) primarily focus on using structured data for Prompt Engineering or Supervised Fine-Tuning (SFT). To better address your concern and highlight the distinctiveness and necessity of our approach, we have conducted additional analyses and experiments.
>
> We have resubmitted the **revised version** of the paper, with all **modifications highlighted in blue**, and added the supplementary experiments into **Section 3.3 Analysis** and **Appendix B More Experimental Results**. We recommend referring to the revised manuscript for more detailed explanations.
>
> ## **1. Environment Training vs. Context Prompting: Comparison with Tool-using Agents**
>
> To demonstrate the necessity of constructing a *Structured Reasoning Environment* versus simply using *Static Reasoning Context*, we compared our method with the **ToG** baseline [1], a strong agentic method that utilizes structured data as external tools and context. The results for GPT-3.5+ToG and GPT-4+ToG are taken from the original ToG paper.
>
> | Methods | WebQSP | CWQ | GrailQA |
> | :--- | :---: | :---: | :---: |
> | **Qwen2.5-7B-Instruct** + SIE-100% + CoT | 26.3 | 34.4 | 40.5 |
> | **Qwen2.5-7B-Instruct** + RL w/ SIE-100% | **93.4** | **87.8** | **85.8** |
> | **Qwen2.5-7B-Instruct** + RL w/ SIE-50% | 86.4 | 78.2 | 81.7 |
> | **Qwen2.5-7B-Instruct** + RL w/ SIE-0% | 72.8 | 56.1 | 71.5 |
> | **Qwen2.5-7B-Instruct** + ToG | 32.1 | 26.0 | 15.3 |
> | **GPT-3.5** + ToG | 76.2 | 57.1 | 68.7 |
> | **GPT-4** + ToG | 82.6 | 67.6 | 81.4 |
>
> The comparison with the tool-using agent reveals three key insights regarding the efficacy of our method. First, the **limitations of static context** are evident: the ToG method, which relies on the model's intrinsic instruction-following and planning skills, performs well with GPT-4 but fails significantly with 7B-scale models (e.g., Qwen2.5-7B-Instruct+ToG achieves only 32.1% on WebQSP), indicating that simply providing structured data as context is insufficient for smaller models. Second, the **superiority of environment training** is clear, as our RL w/ SIE method enables a 7B model to achieve a qualitative leap in structured reasoning (93.4% on WebQSP). Third, and most remarkably, the **internalization of reasoning** is demonstrated by the fact that even in resource-constrained environments like **SIE-50%** (50% supporting facts) or **SIE-0%** (no supporting facts, only distractors), the RL-trained 7B model **outperforms or matches** GPT-3.5+ToG, which proves that SIE is not merely context augmentation but utilizes RL to internalize the capability of exploring and reasoning within structured data into the model parameters.
>
> ## **2. Generalization Across Structure Types: From KGs to Tables (TabMWP)**
>
> To address the generality of the structured data concept and verify that SIE cultivates general structured reasoning skills rather than overfitting to KGs, we added a **Zero-shot generalization** experiment on the tabular question-answering dataset **TabMWP** [2].
>
> | Method | Qwen2.5-7B | Qwen2.5-7B-Instruct | Llama3.1-8B-Instruct |
> | :--- | :---: | :---: | :---: |
> | **CoT** | 45.5 | 41.8 | 69.5 |
> | **RL w/ SIE-100%** | 81.8 (+36.3) | 90.5 (+48.7) | 77.0 (+7.5) |
>
> The results demonstrate the framework's strong **cross-structure generalization** capability: models trained solely in KG-constructed SIE environments can directly transfer to completely different structured data forms (tables). In the zero-shot setting, this transfer yielded **significant performance gains**, resulting in a substantial average performance improvement of over $30.8\%$ across different base models. This confirms that the SIE framework leverages the form of structured data to build a general reasoning environment, effectively enhancing the model's comprehensive understanding and compositional reasoning capabilities for structured information, rather than merely memorizing KG triple patterns.
>
> > [1] Sun J, Xu C, Tang L, et al. Think-on-Graph: Deep and Responsible Reasoning of Large Language Model on Knowledge Graph[C]//ICLR 2024.
> >
> > [2] Lu P, Qiu L, Chang K W, et al. Dynamic Prompt Learning via Policy Gradient for Semi-structured Mathematical Reasoning[C]//ICLR 2023.
>
> We hope the above response addresses your concerns. If our reply has resolved your doubts, maybe you can consider increasing your **Rating** score. If you have any further questions, please let us know promptly. Thank you!

---

> ### Comment · Reviewer_Y1pk · 2025-11-25
>
> Thanks for the response. They make sense to me. I have raised my score.

---

> > ### Author Response · Authors · 2025-11-28
> > **Further Response to Reviewer Y1pk**
> >
> > We greatly appreciate your support and your decision to raise the score. We are pleased that the clarifications and new results were helpful. We will integrate these discussions into the final version to strengthen the manuscript. We welcome any further discussion you might have.

---

### Official Review · Reviewer_7Qwj · 2025-11-01

**Soundness:** 3
**Presentation:** 3
**Contribution:** 2
**Rating:** 4
**Confidence:** 4

**Summary:**

The paper proposes building RL fine-tuning "environments" by packaging KG subgraphs into the prompt, then training LLMs with GRPO using an exact-match/verifiable reward. Construction pulls a seed subgraph from both the question and the answer entities, extracts shortest-path “support” and varies difficulty by hiding support triples. On benchmarks like WebQSP and GrailQA, SIE-RL shows huge gains over the non RL'd baseline and strong OOD benefits on GSM8K and logic puzzles is also reported.

**Strengths:**

1. Interesting concept of creating novel views of standard KGs to RL fine-tune LLMs.
2. Strong results over baselines (non finetuned) are interesting to see.

**Weaknesses:**

1. The intro describes the recipe in too abstract a manner and hypes/overstates the generality of the approach. It is not really an "environment" in the sense of RL and MDPs. Significant LLM usage is acknowledged, however.
2. Reranking with a cross-encoder risks leakage.
3. Huge "SIE-0%" accuracy values suggest that largely using the model’s background knowledge is sufficient to ace the benchmarks, undermining the centrality of the environment. Output format adherence could also explain OOD results. Could this be ablated/tested in isolation somehow?
4. The evaluation setup bakes in supervision (answer-side retrieval) and potentially format adherence for small models that could significantly explain differences in the results.
5. Lack of results for larger models. Only models up to 8b are used where fine-tuning can significantly improve performance. Comparisons also omit strong tool-using KGQA agents.

**Questions:**

1. For the re-ranker cross-encoder leakage risk, could you report results with/without reranking? Also maybe use structure-only distractors?
2. There is also risk for contamination for SRD via the Deepseek R1 API. What do you think?

---

> ### Author Response · Authors · 2025-11-22
> **Response to Reviewer 7Qwj (Part 1)**
>
> We thank the reviewer for the insightful comments. We have addressed your concerns regarding the environment definition, potential leakage risks, ablation studies, and scalability with detailed responses and additional experiments.
>
> We have resubmitted the **revised version** of the paper, with all **modifications highlighted in blue**, and added the supplementary experiments into **Section 3.3 Analysis** and **Appendix B More Experimental Results**. We recommend referring to the revised manuscript for more detailed explanations.
>
> ## **1. Clarification on Environment Definition and SIE Positioning**
>
> We fully agree with the reviewer that SIE is not an explicit environment with immediate state transition feedback as in traditional RL. **As described in Section 2.1**, we frame SIE as an **Implicit MDP** within the LLM reasoning context. In this setting, the LLM’s implicit action is to explore entities and triples within the structured in-context; as reasoning proceeds, the attended subgraph constitutes the updated state; finally, a rule-based verifier provides a scalar reward.
>
> Our core objective is to overcome the limitations of constructing LLM reasoning environments (e.g., lack of scalability and verifiability) by using large-scale structured data to build a scalable, generalizable reasoning sandbox with soft constraints. For future work, we are exploring treating large-scale knowledge graphs as explicit interaction environments to further enhance long-horizon planning of LLMs via Agentic RL.

---

> ### Author Response · Authors · 2025-11-22
> **Response to Reviewer 7Qwj (Part 2)**
>
> ## **2. Analysis of Cross-Encoder Reranking and Potential Leakage Risks**
>
> Regarding concerns about data leakage, it is crucial to distinguish between traditional KGQA tasks (where answer-side information is restricted) and structured environment construction for LLM reasoning. Our core objective is not to optimize KGQA metrics but to leverage massive structured data to automatically construct scalable Structured In-context Environments (SIE), thereby enhancing the general reasoning capabilities of LLMs via RL. Utilizing **answer-side and answer-path** information for environment construction is a common practice in recent works [1-2], aimed at ensuring the completeness and verifiability of reasoning paths in training data. *Crucially, the answer-side information is not explicitly revealed to the model as the answer; instead, it is implicitly embedded within a vast amount of distractor triples as part of the environment.* This construction approach not only ensures high environment quality but also allows precise control over information density (e.g., from SIE-100% to SIE-0%), enabling systematic analysis of the model's exploration and reasoning skills under varying noise levels.
>
> Regarding the use of a cross-encoder for distractor subgraph reranking, we employed a lightweight ms-marco-MiniLM-L12-v2 model (33.4M parameters) solely to compute the semantic similarity between the question and the triples. *Importantly, no answer-side or answer-path information was utilized in this process, ensuring there is no risk of answer leakage via the reranker.* This semantic reranking approach is a widely adopted practice in prior KGQA research to prune entities and relations [3-4].
>
> To further verify the necessity and safety of **semantic reranking**, we conducted additional ablation studies by replacing it with two alternative rerankers: a **random reranker**, which simply retains distractor triples at random, and a **structure reranker**, which retains triples selected through rule-based heuristics that prioritize structural similarity to the supporting subgraph or the presence of entity or relation mentions.
>
> Results based on Qwen2.5-7B-Instruct + RL + SIE-100% are shown below:
>
> | Methods | WebQSP | CWQ | GrailQA | GSM8K | MATH500 | KK-easy | KK-hard |
> | :--- | :---: | :---: | :---: | :---: | :---: | :---: | :---: |
> | **Semantic Reranker** | 93.4 | 87.8 | **85.8** | **87.4** | **61.6** | **49.5** | **29.0** |
> | **Random Reranker** | 93.2 | 87.6 | 84.6 | 87.0 | 61.4 | 49.5 | 26.5 |
> | **Structure Reranker** | **94.9** | **91.2** | 83.8 | 87.1 | 60.2 | 47.0 | 24.5 |
>
> The experimental results demonstrate that the **semantic reranker** (proposed in our paper) achieves **the best overall performance**, particularly in terms of generalization.
>
> Specifically, while the **random reranker** yields comparable results on structured reasoning and simpler math/logic tasks, it exhibits a notable **decline in generalization performance** on the more challenging logical reasoning benchmark (KK-hard, dropping from 29.0% to 26.5%). This suggests that completely random distractors may lack sufficient semantic relevance to establish a challenging reasoning boundary, thereby limiting the model's robustness when handling complex reasoning.
>
> Conversely, the **structure reranker** achieves the highest in-domain scores (e.g., on WebQSP/CWQ) but suffers from the poorest generalization. This indicates that distractors that are too relevant or structurally similar might encourage the model to **exploit structural shortcuts** rather than developing genuine exploration capabilities. Therefore, the semantic reranker represents the optimal trade-off for constructing an environment that fosters generalizable reasoning skills.

---

> ### Author Response · Authors · 2025-11-22
> **Response to Reviewer 7Qwj (Part 3)**
>
> ## **3. Analysis of High SIE-0% Accuracy and Environment Centrality**
>
> Regarding the concern that "high SIE-0% performance undermines the environment's importance," we argue that SIE-0% gains stem from RL activating parametric knowledge and the "negative constraints" provided by distractors. **The substantial gain from SIE-100% is the key to verifying the environment's importance.**
>
> To deeply analyze the role of the environment at different stages, we defined the following four experimental settings:
> - **CoT w/o Context**: A training-free baseline that uses CoT prompting to guide the model to answer directly using its internal knowledge.
> - **RL w/o Context**: RL fine-tuning using only question-answer pairs with rule-based rewards.
> - **RL w/ SIE-0%**: RL fine-tuning where the context contains **only distractor subgraphs** (no supporting subgraphs are retained), along with the question and answer.
> - **RL w/ SIE-100%**: RL fine-tuning where the context contains both the complete supporting subgraph and distractor subgraphs.
>
> In the WebQSP and CWQ datasets, approximately 65% and 40% of the questions are single-hop, respectively. We compare four baselines using Qwen2.5-7B-Instruct.
>
> | Datasets | CoT w/o Context | RL w/o Context | RL w/ SIE-0% | RL w/ SIE-100% |
> | :--- | :---: | :---: | :---: | :---: |
> | **WebQSP** | 2.0 | 59.7 | 72.8 | **93.4** |
> | **CWQ** | 8.2 | 36.7 | 56.1 | **87.7** |
>
> The analysis of the results is as follows:
>
> The improvement of **RL w/o Context over CoT w/o Context** indicates that RL successfully activates the LLM's **internal parametric knowledge** for structured reasoning, primarily resolving simpler single-hop questions.
>
> The gain of **RL w/ SIE-0% over RL w/o Context** suggests that even without supporting subgraphs, the presence of distractors serves as a critical **negative constraint**. This forces the model to learn how to **prune incorrect reasoning paths** based on distractor triples, yielding a performance boost of ~13-20%.
>
> The substantial lift of **RL w/ SIE-100% over RL w/ SIE-0%** demonstrates that the inclusion of supporting subgraphs effectively **extends the model's knowledge boundary**. It teaches the LLM to synthesize its parametric knowledge with external environmental knowledge for **complex multi-hop compositional reasoning**, resulting in an additional ~20-30% improvement, which is a capability that cannot be derived from internal knowledge alone.

---

> ### Author Response · Authors · 2025-11-22
> **Response to Reviewer 7Qwj (Part 4)**
>
> ## **4. Ablation Study on Format Adherence**
>
> To address the concern that format adherence might explain the performance gains and to investigate potential contamination from SRD distilled data, we introduced a **Format Only** baseline. In this setting, we reward the model solely for compliance with the output format, disregarding answer correctness. The results under the Qwen2.5-7B-Instruct + SIE-100% setting are presented below (Note: RL w/ Answer + Format corresponds to RL w/ SIE in the original paper):
>
> | Methods | WebQSP | CWQ | GrailQA | GSM8K | MATH500 | KK-easy | KK-hard |
> | :--- | :---: | :---: | :---: | :---: | :---: | :---: | :---: |
> | **Qwen2.5-7B-Instruct** | 26.3 | 34.4 | 40.5 | 29.2 | 43.0 | 42.0 | 19.5 |
> | &nbsp;&nbsp;&nbsp;&nbsp;+ SFT w/ SRD | 40.5 | 43.3 | 55.7 | 68.1 | 54.8 | 41.5 | 21.5 |
> | &nbsp;&nbsp;&nbsp;&nbsp;+ RL w/ Answer + Format | **93.4** | **87.8** | **85.8** | **87.4** | **61.6** | **49.5** | **29.0** |
> | &nbsp;&nbsp;&nbsp;&nbsp;+ RL w/ Format only | 31.6 | 37.7 | 48.1 | 37.0 | 46.6 | 44.0 | 20.0 |
> | **Llama3.1-8B-Instruct** | 36.5 | 37.2 | 43.6 | 67.4 | 38.4 | 20.5 | 6.0 |
> | &nbsp;&nbsp;&nbsp;&nbsp;+ SFT w/ SRD | 43.4 | 49.5 | 60.0 | 73.6 | 42.0 | 8.5 | 1.5 |
> | &nbsp;&nbsp;&nbsp;&nbsp;+ RL w/ Answer + Format | **93.2** | **89.7** | **85.0** | **82.6** | **47.0** | **37.0** | **15.5** |
> | &nbsp;&nbsp;&nbsp;&nbsp;+ RL w/ Format only | 45.1 | 44.1 | 56.1 | 68.8 | 43.4 | 30.0 | 11.0 |
>
> **Comparison between SFT w/ SRD and RL w/ Answer + Format.** Theoretically, the gains in SFT w/ SRD stem from two sources: format adherence and the imitation of DeepSeek-R1's reasoning paradigms. However, despite benefiting from the distilled knowledge of R1, the SFT baseline significantly underperforms RL w/ Answer + Format. This gap indicates that for 7B-8B scale models, neither mere format adherence nor the prior knowledge from R1 is the dominant factor for performance improvement. The substantial advantage of the RL method strongly demonstrates that reinforcement exploration within a structured environment is far more critical than simple imitative reasoning.
>
> **Comparison between RL w/ Format Only and RL w/ Answer + Format.** The Format Only baseline yields only marginal gains over the CoT baseline (slightly higher for the Llama series than Qwen), which is primarily attributed to standardized outputs facilitating easier answer parsing. However, RL w/ Answer + Format achieves a significant performance leap on top of this. This provides compelling evidence that the gains from RL w/ SIE are fundamentally driven by enhanced reasoning capabilities fostered by the structured environment, rather than mere format adherence.

---

> ### Author Response · Authors · 2025-11-22
> **Response to Reviewer 7Qwj (Part 5)**
>
> ## **5. Experiments with Larger LLMs and Tool-using Agents**
>
> **Scalability to 14B Models**: We added experiments with Qwen2.5-14B-Instruct. The 14B model achieves superior results under RL w/ SIE-100% compared to the 7B model, demonstrating the framework's effectiveness and scalability.
>
> | Methods | WebQSP | CWQ | GrailQA | GSM8K | MATH500 | KK-easy | KK-hard |
> | :--- | :---: | :---: | :---: | :---: | :---: | :---: | :---: |
> | **Qwen2.5-7B-Instruct** | 26.3 | 34.4 | 40.5 | 29.2 | 43.0 | 42.0 | 19.5 |
> | &nbsp;&nbsp;&nbsp;&nbsp;+ RL w/ SIE-100% | 93.4 | 87.8 | 85.8 | 87.4 | 61.6 | 49.5 | 29.0 |
> | **Qwen2.5-14B-Instruct** | 40.9 | 48.0 | 65.6 | 72.1 | 62.4 | 60.5 | 35.0 |
> | &nbsp;&nbsp;&nbsp;&nbsp;+ RL w/ SIE-100% | **94.0** | **89.9** | **87.4** | **91.1** | **75.0** | **66.0** | **45.5** |
>
> **Comparison with Tool-using Agents (ToG)**: We compared our method with a strong ToG baseline [3]. The results for GPT-3.5+ToG and GPT-4+ToG are taken from the original ToG paper.
>
> | Methods | WebQSP | CWQ | GrailQA |
> | :--- | :---: | :---: | :---: |
> | **Qwen2.5-7B-Instruct** + SIE-100% + CoT | 26.3 | 34.4 | 40.5 |
> | **Qwen2.5-7B-Instruct** + RL w/ SIE-100% | **93.4** | **87.8** | **85.8** |
> | **Qwen2.5-7B-Instruct** + RL w/ SIE-50% | 86.4 | 78.2 | 81.7 |
> | **Qwen2.5-7B-Instruct** + RL w/ SIE-0% | 72.8 | 56.1 | 71.5 |
> | **Qwen2.5-7B-Instruct** + ToG | 32.1 | 26.0 | 15.3 |
> | **GPT-3.5** + ToG | 76.2 | 57.1 | 68.7 |
> | **GPT-4** + ToG | 82.6 | 67.6 | 81.4 |
>
> Results show that while **ToG** performs well with GPT-3.5/4, it **struggles with 7B models**. In contrast, SIE significantly boosts small model reasoning. Even in partial settings (**SIE-50% or SIE-0%**), the RL-trained 7B model **exceeds or matches GPT-3.5+ToG**, proving SIE's effectiveness in enhancing structured reasoning for smaller LLMs.
>
> > [1] Lu R, Hou Z, Wang Z, et al. Deepdive: Advancing deep search agents with knowledge graphs and multi-turn rl[J]. arXiv preprint arXiv:2509.10446, 2025.
> >
> > [2] Yu A, Yao L, Liu J, et al. Medresearcher-r1: Expert-level medical deep researcher via a knowledge-informed trajectory synthesis framework[J]. arXiv preprint arXiv:2508.14880, 2025.
> >
> > [3] Sun J, Xu C, Tang L, et al. Think-on-Graph: Deep and Responsible Reasoning of Large Language Model on Knowledge Graph[C]//ICLR 2024.
> >
> > [4] Chen L, Tong P, Jin Z, et al. Plan-on-graph: Self-correcting adaptive planning of large language model on knowledge graphs[J]. NeurIPS 2024.
>
> We hope the above response addresses your concerns. If our reply has resolved your doubts, maybe you can consider increasing your **Rating** score. If you have any further questions, please let us know promptly. Thank you!

---

> > ### Comment · Reviewer_7Qwj · 2025-11-27
> >
> > Thanks for the follow up. My concerns are addressed and I have increased the score

---

> > > ### Author Response · Authors · 2025-11-28
> > > **Further Response to Reviewer 7Qwj**
> > >
> > > We are sincerely grateful for your valuable feedback and your decision to increase the score. We are glad that our response addressed your concerns. We will incorporate the additional experiments and analyses into the final manuscript to further improve the paper. We remain open to any further discussion.

---

### Official Review · Reviewer_YQJe · 2025-11-02

**Soundness:** 2
**Presentation:** 3
**Contribution:** 3
**Rating:** 6
**Confidence:** 3

**Summary:**

The paper proposes the Structured In-context Environment (SIE) framework, which builds scalable, verifiable reasoning environments for RL post-training directly from large structured data (instantiated with knowledge graphs). SIE is inserted as a structured context in the prompt—so the model’s “actions” are its in-context explorations—and rewards are rule-based (exact-match answer and formatting), optimized with GRPO (and also tested with PPO/REINFORCE++). An automated pipeline retrieves a compact seed subgraph, extracts shortest-path “support” subgraphs, semantically filters distractors, and constructs partial SIEs by controlling the retained support ratio while fixing context length. Across Qwen and Llama backbones, RL in SIE yields large gains on structured reasoning benchmarks (WebQSP, CWQ, GrailQA) over CoT, SFT on distilled structured reasoning data, and RL without context, and the learned compositional skills transfer to out-of-domain math and logic tasks.

**Strengths:**

The paper is original in how it reframes reasoning RL as an environment-design problem: it turns large structured data into *Structured In-context Environments (SIEs)* so that reasoning happens via in-context exploration with verifiable, rule-based rewards, rather than only via prompts or dataset distillation. The automated pipeline—seed subgraph retrieval, shortest-path “support” extraction, semantic distractor filtering, and construction of *partial* SIEs that control information content—both grounds the idea technically and makes it scalable; The empirical quality is strong and maintains sizable advantages even when support evidence is aggressively removed, demonstrating robustness to incomplete information. The significance extends beyond KGQA: the learned strategies transfer to out-of-domain math and logic tasks, indicating that SIEs elicit compositional skills rather than overfitting to a specific graph schema.

**Weaknesses:**

The out-of-domain math evaluation focuses on GSM8K and MATH500, which leaves open whether the learned strategies carry to the current hardest math leaderboards; adding AIME 2024/2025 and Olympiad-style sets would provide a more stringent test of compositional reasoning under scarce-signal regimes.  Although the paper includes Llama3.1-8B-Instruct both in-domain and for OOD transfer, the results are not shown;  Finally, given recent concerns that Qwen-family models can improve under weak or even randomized reward shaping, the study would be stronger with explicit controls: reward-shuffled and random-reward baselines.

**Questions:**

see more details in the weakness.

---

> ### Author Response · Authors · 2025-11-22
> **Response to Reviewer YQJe (Part 1)**
>
> We thank the reviewer for the constructive feedback. We have conducted additional experiments to address your concerns regarding generalization to harder mathematical benchmarks, the presentation of Llama3.1 results, and the control studies for reward mechanisms.
>
> We have resubmitted the **revised version** of the paper, with all **modifications highlighted in blue**, and added the supplementary experiments into **Section 3.3 Analysis** and **Appendix B More Experimental Results**. We recommend referring to the revised manuscript for more detailed explanations.
>
> ## **1. Generalization Evaluation on AIME 2024/2025 and TabMWP**
>
> To verify whether the learned strategies generalize to scarce-signal and highly challenging regimes, we further evaluated our method (RL w/ SIE-100%) on **AIME 2024/2025** (rigorous Olympiad-level math benchmarks) and **TabMWP** [1] (a table-based structured question-answering dataset). The inclusion of TabMWP aims to verify whether the structured reasoning capabilities learned from KGs can transfer to other forms of structured data, such as tables.
>
> The results are shown in the table below (reporting pass@k with 16 samples for AIME):
>
> | Methods | AIME 24 pass@1 | AIME 24 pass@2 | AIME 24 pass@4 | AIME 24 pass@8 | TabMWP accuracy |
> | :--- | :---: | :---: | :---: | :---: | :---: |
> | **Qwen2.5-7B** | 2.29 | 3.97 | 6.27 | 9.22 | 45.5 |
> | &nbsp;&nbsp;&nbsp;&nbsp;+ RL w/ SIE-100% | 6.25 (+3.96) | 10.31 (+6.34) | 15.02 (+8.75) | 19.41 (+10.19) | 81.8 (+36.3) |
> | **Llama3.1-8B-Instruct** | 3.12 | 4.97 | 7.31 | 10.89 | 69.5 |
> | &nbsp;&nbsp;&nbsp;&nbsp;+ RL w/ SIE-100% | 4.58 (+1.46) | 7.81 (+2.84) | 12.12 (+4.81) | 17.30 (+6.41) | 77.0 (+7.5) |
>
> | Methods | AIME 25 pass@1 | AIME 25 pass@2 | AIME 25 pass@4 | AIME 25 pass@8 |
> | :--- | :---: | :---: | :---: | :---: |
> | **Qwen2.5-7B** | 1.46 | 2.83 | 5.36 | 9.67 |
> | &nbsp;&nbsp;&nbsp;&nbsp;+ RL w/ SIE-100% | 3.54 (+2.08) | 6.67 (+3.84) | 11.83 (+6.47) | 18.87 (+9.20) |
> | **Llama3.1-8B-Instruct** | 0.42 | 0.83 | 1.67 | 3.33 |
> | &nbsp;&nbsp;&nbsp;&nbsp;+ RL w/ SIE-100% | 0.63 (+0.21) | 1.22 (+0.39) | 2.33 (+0.66) | 4.22 (+0.89) |
>
> On the extremely difficult AIME 2024/2025 benchmarks, our method generally demonstrates **stable performance advantages** over the CoT baseline as $k$ increases. This indicates that models trained within SIE possess stronger exploration capabilities and robustness when dealing with complex, multi-step math reasoning tasks.
>
> Moreover, on TabMWP, our method achieved a substantial average performance improvement of **21.9%** in the zero-shot setting. This confirms that the reasoning capabilities cultivated by the SIE framework are not limited to KG structures but can effectively transfer to heterogeneous structured data like tables, demonstrating genuine structural reasoning generalization.
>
> ## **2. In-Domain and Out-of-Domain Results for Llama3.1-8B-Instruct**
>
> We respectfully clarify that, due to space limitations, the full experimental results for Llama3.1-8B-Instruct were included in **Appendix B (Tables 10 and 11)** of our **revised submission** and in Appendix B (Tables 8 and 9) of the original submission. For your convenience, we summarize the key results below. Here, *SFT w/ SRD* refers to the supervised fine-tuning baseline using distilled Chain-of-Thought reasoning data from DeepSeek-R1.
>
> | Methods | GrailQA | GSM8K | MATH500 | KK-easy | KK-hard |
> | :--- | :---: | :---: | :---: | :---: | :---: |
> | **CoT** | 43.6 | 67.4 | 38.4 | 20.5 | 6.0 |
> | **SFT w/ SRD** | 60.0 (+16.4) | 73.6 (+6.2) | 42.0 (+3.6) | 8.5 (-12.0) | 1.5 (-4.5) |
> | **RL w/ SIE-100%** | **85.0** (+41.4) | **82.6** (+15.2) | **47.0** (+8.6) | **37.0** (+16.5) | **15.5** (+9.5) |
>
> The results clearly demonstrate that our *RL w/ SIE* method is equally effective on Llama3.1-8B-Instruct, with generalization performance far exceeding the SFT w/ SRD baseline. Notably, while *SFT w/ SRD* improves performance on some tasks, it causes significant *degradation* (negative transfer) on logical reasoning tasks (KK-easy and KK-hard). In contrast, RL w/ SIE achieves consistent and significant improvements across all in-domain and out-of-domain tasks, demonstrating the universality and superiority of the SIE framework.

---

> ### Author Response · Authors · 2025-11-22
> **Response to Reviewer YQJe (Part 2)**
>
> ## **3. Ablation Study: Format-Only vs. Random Rewards**
>
>
> To explicitly address the concern that Qwen models might improve under weak or randomized reward shaping [2], we introduced two control baselines under the RL w/ SIE-100% setting:
> - **Format Only**: Removes the correctness reward, rewarding the model solely for following the \<think\> and \<answer\> response format.
> - **Random + Format**: Replaces the answer reward with a random 0 or 1 reward (50% probability), while retaining the format reward.
>
> | Methods | WebQSP | CWQ | GrailQA | GSM8K | MATH500 | KK-easy | KK-hard |
> | :--- | :---: | :---: | :---: | :---: | :---: | :---: | :---: |
> | **Qwen2.5-7B-Instruct** | 26.3 | 34.4 | 40.5 | 29.2 | 43.0 | 42.0 | 19.5 |
> | &nbsp;&nbsp;&nbsp;&nbsp;+ Answer + Format | **93.4** | **87.8** | **85.8** | **87.4** | **61.6** | **49.5** | **29.0** |
> | &nbsp;&nbsp;&nbsp;&nbsp;+ Format Only | 31.6 | 37.7 | 48.1 | 37.0 | 46.6 | 44.0 | 20.0 |
> | &nbsp;&nbsp;&nbsp;&nbsp;+ Random + Format | 6.3 | 6.4 | 6.7 | 12.6 | 26.6 | 40.5 | 19.0 |
> | **Llama3.1-8B-Instruct** | 36.5 | 37.2 | 43.6 | 67.4 | 38.4 | 20.5 | 6.0 |
> | &nbsp;&nbsp;&nbsp;&nbsp;+ Answer + Format | **93.2** | **89.7** | **85.0** | **82.6** | **47.0** | **37.0** | **15.5** |
> | &nbsp;&nbsp;&nbsp;&nbsp;+ Format Only | 45.1 | 44.1 | 56.1 | 68.8 | 43.4 | 30.0 | 11.0 |
> | &nbsp;&nbsp;&nbsp;&nbsp;+ Random + Format | 37.7 | 39.7 | 52.6 | 66.9 | 42.8 | 27.0 | 8.5 |
>
> Based on these results, we can draw the following conclusions:
>
> **Format reward is not the primary driver**: While the *Format Only* baseline yields marginal gains (slightly higher for Llama than Qwen), primarily due to more standardized output facilitating answer extraction, the magnitude of improvement is far less than our proposed method (Answer + Format). This indicates that the qualitative leap in performance stems from sufficient exploration based on correct feedback within the structured in-context environment, not merely learning the response format.
>
> **Random rewards cause performance collapse**: When random rewards are introduced, the performance of the Qwen series (which has strong instruction-following capabilities) drops sharply, falling significantly below even the CoT baseline. This suggests the model is not cheating with random signals but is instead severely misled by incorrect rewards. While the Llama series retains some performance under random rewards, it still underperforms the Format Only baseline. This proves that random reward signals are detrimental to structured in-context reasoning.
>
> These results confirm that the significant improvements achieved by the SIE framework are derived from the model truly learning compositional reasoning paradigms guided by correct, verifiable rewards.
>
> > [1] Lu P, Qiu L, Chang K W, et al. Dynamic Prompt Learning via Policy Gradient for Semi-structured Mathematical Reasoning[C]//ICLR 2023.
> >
> > [2] Shao R, Li S S, Xin R, et al. Spurious rewards: Rethinking training signals in rlvr[J]. arXiv preprint arXiv:2506.10947, 2025.
>
> We hope the above response addresses your concerns. If our reply has resolved your doubts, maybe you can consider increasing your **Rating** score. If you have any further questions, please let us know promptly. Thank you!

---

### Author Response · Authors · 2025-12-01
**Summary of Rebuttal and Revision Updates**

Dear Area Chair,

We have provided comprehensive responses and conducted additional experiments to address all reviewers' concerns. **We are pleased to report that Reviewer 7Qwj and Reviewer Y1pk have both acknowledged our efforts, stating that their concerns have been resolved, and have raised their ratings to 6**. **Prior to the score reverting, our average rating was a solid 6**. We believe the current version of the paper is significantly stronger and validates the effectiveness of our approach.

**Summary of Revision:**

We have uploaded a revised manuscript with **all changes highlighted in blue**. The major revisions are as follows:

* **Expansion of Section 3.3 (Analysis):** We added two key subsections with supporting tables (**Table 8 & 9**) to analyze the impact of reward mechanisms and decompose the sources of reasoning improvements. These analyses explicitly rule out format adherence or reward gaming as drivers of performance and quantify the value of the structured in-context environment.

* **Enrichment of Appendix B (More Experimental Results):** To further strengthen the empirical robustness and quality of our work, we added four new subsections (**B.2–B.5**) containing extensive supplementary experiments. These additions include generalization evaluations on rigorous benchmarks (AIME 2024 and TabMWP), ablation studies on distractor reranking strategies (Semantic vs. Random vs. Structure), scalability verification using the larger Qwen2.5-14B-Instruct model, and a comparative analysis with Tool-using Agents (ToG), all of which provide comprehensive evidence for the efficacy of the proposed framework.

**Response Summaries:**

* **To Reviewer YQJe:** We addressed concerns about **generalization and reward mechanisms**. We added experiments on the rigorous AIME 2024/2025 and TabMWP benchmarks, showing that our method achieves stable gains in complex math and zero-shot transfer to tabular data (+21.9% accuracy). We also clarified that full Llama3.1 results were in the appendix and show strong performance. Crucially, our reward ablation study (Format Only vs. Random Rewards) proved that performance gains stem from learning correct reasoning logic, not gaming the format or random signals.

* **To Reviewer 7Qwj:** We clarified the environment definition (Implicit MDP) and addressed data leakage concerns. We explained that using answer-side info for environment construction is standard and safe, as confirmed by our reranker ablation study (Semantic vs. Random vs. Structure), which showed our semantic reranker balances difficulty and generalization best without leakage. We further demonstrated scalability with Qwen2.5-14B-Instruct and superiority over Tool-using Agents (ToG), where our 7B model outperforms GPT-3.5+ToG even in information-limited settings. **Reviewer 7Qwj confirmed these points addressed their concerns and raised their score.**

* **To Reviewer Y1pk:** We distinguished our work from standard context prompting by emphasizing the construction of scalable RL environments. We provided a direct comparison with ToG, showing that while static context fails for small models, our environment training enables qualitative leaps in reasoning (e.g., 93.4% vs 32.1% on WebQSP). We also demonstrated cross-structure generalization to tabular data (TabMWP), proving our method cultivates general structural reasoning skills rather than overfitting to KGs. **Reviewer Y1pk found these responses sensible and raised their score.**

Thank you for your time and consideration.

Best regards,
The Authors of Submission No.9144

---

### Meta-Review · Area_Chair_W3Nk · 2026-01-05

**Summary:**

The following concerns were raised during the reviews:

C1. The out-of-domain math evaluation targets GSM8K and MATH 500. More challenging datasets should also be included.

C2. There should be a comparison to reward-shuffled and random-reward baselines.

C3. The guidance towards format adherence for small models could significantly explain the differences in the results.

C4. The results are restricted to small models.

**Reviewer Concerns:**

All of the concerns were addressed during the rebuttal and the reviewers acknowledged that they were satisfied by the responses and would raise their scores.

**Reviewer Scores:**

Reviewer YQJe. POSSIBLE INCREASE. The reviewer already awarded a 6. The response addressed the specified concerns, so there is a reasonable chance of an increase of the score.

Reviewer 7QWj. RAISED to 6 (indicated in response).

Reviewer Y1pk. RAISED to 6 (indicated in response).

---

### Decision · Program_Chairs · 2026-01-26

Accept (Poster)